# Rational strain engineering of single-atom ruthenium on nanoporous MoS$_2$ for highly efficient hydrogen evolution

Kang Jiang[1], Min Luo[2], Zhixiao Liu[1], Ming Peng[1], Dechao Chen[1], Ying-Rui Lu[3], Ting-Shan Chan [3], Frank M. F. de Groot[4] & Yongwen Tan [1✉]

Maximizing the catalytic activity of single-atom catalysts is vital for the application of single-atom catalysts in industrial water-alkali electrolyzers, yet the modulation of the catalytic properties of single-atom catalysts remains challenging. Here, we construct strain-tunable sulphur vacancies around single-atom Ru sites for accelerating the alkaline hydrogen evolution reaction of single-atom Ru sites based on a nanoporous MoS$_2$-based Ru single-atom catalyst. By altering the strain of this system, the synergistic effect between sulphur vacancies and Ru sites is amplified, thus changing the catalytic behavior of active sites, namely, the increased reactant density in strained sulphur vacancies and the accelerated hydrogen evolution reaction process on Ru sites. The resulting catalyst delivers an over-potential of 30 mV at a current density of 10 mA cm$^{-2}$, a Tafel slope of 31 mV dec$^{-1}$, and a long catalytic lifetime. This work provides an effective strategy to improve the activities of single-atom modified transition metal dichalcogenides catalysts by precise strain engineering.

[1] College of Materials Science and Engineering, State Key Laboratory of Advanced Design and Manufacturing for Vehicle Body, Hunan University, Changsha, Hunan 410082, China. [2] Department of Physics, Shanghai Polytechnic University, Shanghai 201209, China. [3] National Synchrotron Radiation Research Center, Hsinchu 300, Taiwan. [4] Inorganic Chemistry & Catalysis, Debye Institute for Nanomaterials Science, Utrecht University, Universiteitsweg 99, 3584 CG Utrecht, The Netherlands. ✉email: tanyw@hnu.edu.cn

The design of high-performance and cost-effective heterogeneous catalysts for the hydrogen evolution reaction (HER) is critical for the development of efficient water electrolyzers[1,2]. Single-atom catalysts (SACs) were considered as ideal HER electrocatalysts for achieving high catalytic activity and reducing the metal loading due to their maximized atom-use efficiency, well-defined single-atom dispersion, and unique coordination environments[3–7]. Nevertheless, the catalytic activity of state-of-the-art SACs still has plenty of room for improvement with the aim of maximizing the catalytic activity, especially for multistep reactions (such as carbon dioxide reduction reaction, oxygen reduction reaction, and alkaline HER)[8–12]. This limitation arises from the simplicity of the single-atom sites which are generally capable of efficiently catalyzing one step of reactions rather than whole reactions. Although recently reported catalysts with dual sites (dual metal-atom or metal-atom coordinated with non-metal atom) have achieved the enhanced catalytic activity through synergistic interaction between the dual sites[13–15], it remains a great challenge to unveil the catalytic process over multi-atom sites because of the difficulties in atomically precise preparations. Therefore, constructing a synergistic site to assist the single-atom site may be a promising approach to further enhance the catalytic performance of SACs. Considering these views, it is vital to find a supported material which not only can stabilize the isolated metal atoms, but also can in situ construct the assisting sites around the single-atom sites.

As a typical cost-effective layered transition metal dichalcogenide, molybdenum disulfide ($MoS_2$) has been extensively studied for the HER, where one guiding principle is to activate the inert basal plane sites[16,17]. A variety of strategies, such as phase engineering[18–20], creating sulfur vacancies (SVs)[21,22], and single-atom doping[23–27] have been successively invented to activate the basal plane. The introduction of single-atom could create SVs in the $MoS_2$ basal plane[23]. The SVs around single-atom are generally considered as the active site for HER, but deep insight into the synergetic effect of SVs on single-atom has not be achieved, especially under realistic reaction conditions. Thus, it is highly desirable to find a way to amplify the synergetic interaction between SVs and single-atoms and mechanistically understand the synergetic effect, thus maximizing the catalytic activity of SACs. Interestingly, introducing strain into catalysts can optimize the electronic structure of active sites (single-atom and SV)[21,28,29], thus creating reaction-favorable environment for reactant, which may be a robust strategy for enhancing the intrinsic HER activity.

Inspired by this, we set out to construct a nanoporous $MoS_2$ (denoted as np-$MoS_2$) with bicontinous structure to anchor single-atom Ru (denoted as Ru/np-$MoS_2$) and selected alkaline HER as a model reaction to explore the synergistic effect between Ru sites and SVs. The curvature-induced strain can be precisely tailored by tuning the ligament size of nanoporous $MoS_2$ (Fig. 1a). The best catalyst, Ru/np-$MoS_2$, delivers an overpotential of 30 mV to achieve a current density of 10 mA cm$^{-2}$ and a Tafel slope of 31 mV dec$^{-1}$ for alkaline HER. By using theoretical calculations, operando X-ray absorption spectroscopy (XAS), and ambient pressure X-ray photoelectron spectroscopy (AP-XPS) techniques, it is identified that the applied strain can enhance the accumulation of OH$^-$ and $H_2O$ in SVs resulting in the increase of reactant density in the inner Helmholtz plane, thus accelerating the mass transfer to Ru sites. Simultaneously, the bending strain of the Ru/np-$MoS_2$ effectively modulates the electronic structure of single-atom Ru, which could catalyze the $H_2O$ dissociation and H−H coupling more effectively. This work provides atomic-level insight into the SVs-synergetic effect for single-atom Ru sites and amplifies this effect by the introduction of strain, which will be helpful in designing high active catalysts.

## Results

**Theoretical calculations.** Herein, we selected Ru as the single-atom sites for constructing this system because of its apparent performances for HER[12]. In light of previous reports regarding the activation of $MoS_2$ basal plane by the doping of isolated metal atoms[23], we hypothesized that the introduction of isolated Ru atoms into $MoS_2$ could cause the loss of S atoms around Ru atoms, accompanied with phase conversion to form Ru/1T-$MoS_2$. The formation of SVs could break the steric effect and allow the direct binding between Mo atoms and $H_2O$ molecule in SVs (Fig. 1a). This hypothesis was suggested by experimental observations in ref. [23]. The formation energy of the Ru atom replacing the Mo site was calculated (Supplementary Fig. 1). It is shown that Ru exhibits a tendency to replace Mo with an exothermic energy of −0.650 eV, indicating the substitutional doping of Ru is a thermodynamically-driven process. Then, we calculated the formation energy of SVs in 1T-$MoS_2$ and Ru/1T-$MoS_2$, which show the decrease in the formation energy of SVs by 0.832 eV after Ru doping, proving the feasibility of using Ru doping to create SVs. Note that the Mo sites located below the SVs are the active sites of SVs (denoted as Mo$_{SV}$)[21]. Therefore, density functional theory (DFT) was employed to assess the role of Ru and Mo$_{SV}$ sites in the HER process (Supplementary Note 1). Next, we investigated the effect of tensile strain on Ru and Mo$_{SV}$ sites (Fig. 1b–d, Supplementary Figs. 2–5, and Supplementary Table 1). We first examined water adsorption energies ($\Delta E_{H2O}$) on various sites of Ru/$MoS_2$ without strain. The results clearly show that $H_2O$ molecules on the Ru sites of Ru/$MoS_2$ possess much lower water adsorption energy of −0.516 eV as compared to that of Mo$_{SV}$ sites, which suggests that the initial $H_2O$ molecules can readily adsorb on the Ru sites (Supplementary Fig. 2). Simultaneously, Ru sites could easily activate $H_2O$ molecule to generate intermediate H and OH species due to its low energy barriers of Volmer step ($\Delta G(H_2O)$) (Supplementary Fig. 3). The subsequently H−H coupling can be completed by Ru sites, resulting from their small hydrogen adsorption free energy ($\Delta G(H^*)$) (Fig. 1d).

After the applied strain, the Mo$_{SV}$ sites display much lower water adsorption energy, indicating the Mo$_{SV}$ sites bear a strong water affinity (Supplementary Fig. 2). The projected density of states (PDOS) of *OH$_2$ (Fig. 1c) further shows that the Mo 3$d$ orbitals and the O$_{ads}$ (the O atom directly bonded to the surface) 2$p$ orbitals have more overlap below the Fermi level after the applied strain, suggesting the stronger binding between *OH$_2$ and Mo$_{SV}$ sites. Therefore, the Mo$_{SV}$ sites may play a role of reactant ($H_2O$) dragging thus enhancing the mass transfer of $H_2O$ to Ru sites. The applied strain also leads to the enhanced DOS of Ru 3$d$ orbitals near the Fermi level, indicating the improvement of $d$-electron domination. Benefiting from this, the energy barriers of Volmer step for Ru sites decrease after the applied strain, leading to the more rapid Volmer step. Besides, the application of strain also decreases the hydrogen adsorption free energy for Ru sites and S sites, thus leading to the enhanced ability for H−H coupling. These predicted the synergistic effect between single-atom Ru sites and SVs, which could be amplified by the introduction of strain in $MoS_2$.

**Synthesis and characterization of catalysts.** Inspired by the theoretical predictions, a np-$MoS_2$ was synthesized via chemical vapor deposition and chemical etching method, which possesses the three-dimensional bicontinuous nanoporous structure and nanotube-shaped ligaments (Fig. 1a, Methods, and Supplementary Figs. 6, 7)[30,31]. Then, SVs were created with the introduction of isolated Ru atoms in the basal plane of $MoS_2$, forming the desired catalyst (Ru/np-$MoS_2$) through a spontaneous reduction

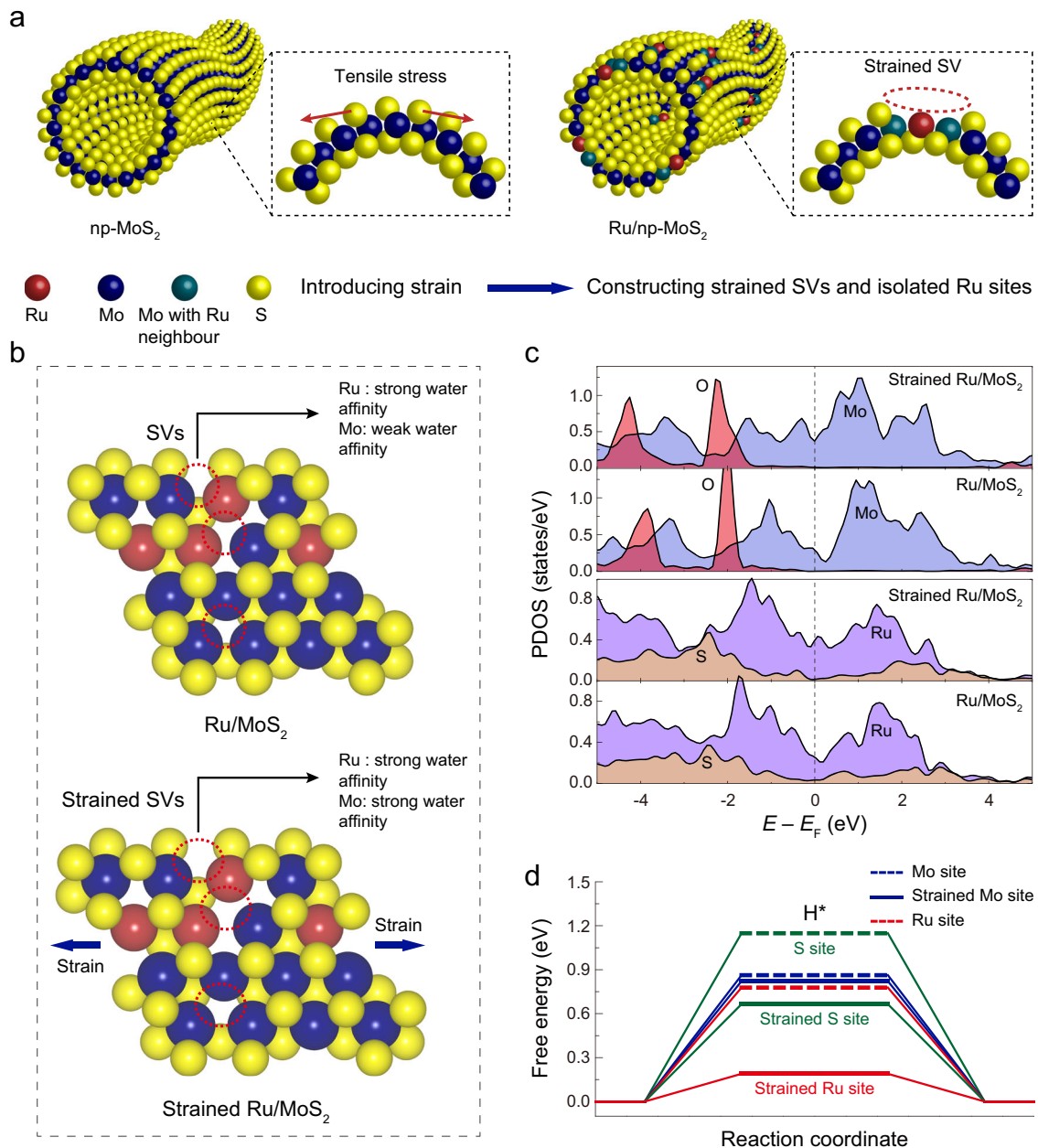

**Fig. 1 DFT calculations for the strain effects. a** Illustration of the construction Ru/np-MoS₂. **b** Geometries of Ru/MoS₂ before and after the applied strain. **c** Calculated PDOS of Ru/MoS₂ before and after the applied strain. **d** Free energy diagrams for hydrogen adsorption at different sits.

strategy[23,32]. Representative scanning electron microscopy (SEM) and transmission electron microscopy (TEM) images emphasize the three-dimensional bicontinuous nanoporous morphology of the as-prepared Ru/np-MoS₂ (Fig. 2a and Supplementary Figs. 7b, 8), consisted of interconnected nanotube with concave and convex curvatures. High-resolution transmission electron microscopy (HRTEM) image not only reveals the atomically curved MoS₂ from the cross-sectional view, but also indicates that the interconnected nanotubular structure is mainly constructed by high-quality few-atomic thick MoS₂ (Fig. 2b). High-angle annular dark-field scanning transmission electron microscopy (HAADF-STEM) confirms the co-existence of 2H⁻ and 1T-MoS₂ in Ru/np-MoS₂ (Fig. 2c, d). The doping of Ru brings out the 2H⁻1T phase transition in np-MoS₂, as evidenced by the decrease in the white line resonance strength of S K- and Mo L₃-edges X-ray absorption near-edge structure (XANES) spectra for Ru/np-MoS₂ (Fig. 2e and Supplementary

Fig. 9)[23]. The magnified HAADF-STEM image of Ru/np-MoS₂ and corresponding intensity profile analyses reveal the loss of S atoms after the substitutional doping of Ru, thus forming SVs (Fig. 2f). Furthermore, the intensity profile analyses of Ru/np-MoS₂ and plane MoS₂ supported Ru SACs (denoted as Ru/P-MoS₂, Supplementary Note 2 and Supplementary Fig. 10) reveal the existence of tensile strain in Ru/np-MoS₂ (Fig. 2g and Supplementary Fig. 11)[33]. The substitutional doping of Ru is further revealed by the HAADF-STEM image at 1T phase region (Fig. 2h, i). In the spontaneous reduction reaction, the energetic Mo vacancies (denoted as MVs, as highlighted by the red circles) on the surface of MoS₂ provide anchoring sites for Ru species (as highlighted by the white circles). While the reduction of Mo by the electrons injection from Ru leads to the phase transformation of MoS₂ into the 1T structure, accompanied by the formation of SVs. The presence of Ru is also verified by STEM energy-dispersive X-ray (STEM-EDX)

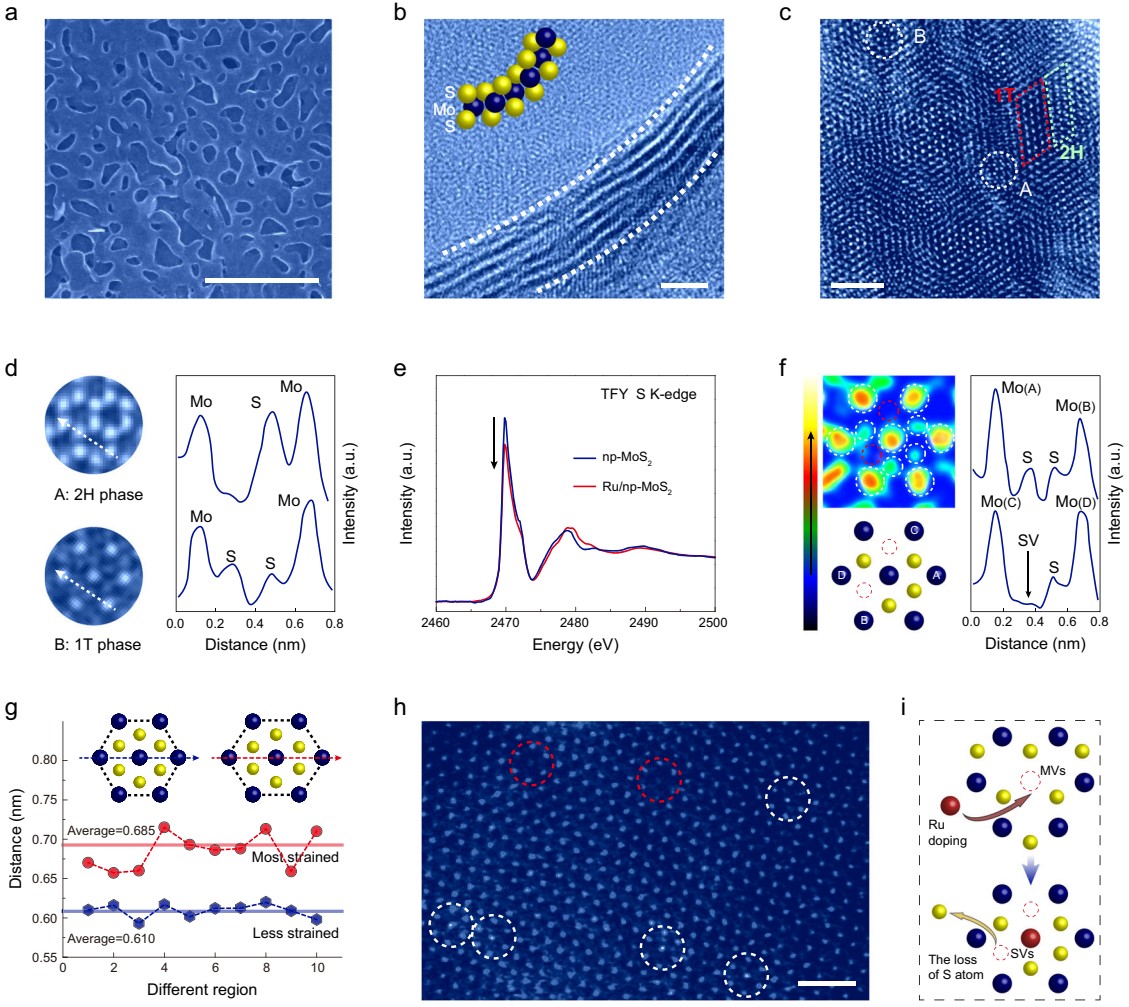

**Fig. 2 Structural characterizations of Ru/np-MoS₂.** **a** SEM image of Ru/np-MoS₂. **b** HRTEM image of Ru/np-MoS₂ from the cross-sectional view. **c** HAADF-STEM image of Ru/np-MoS₂. **d** Magnified HAADF-STEM image and corresponding intensity line profiles. **e** S K-edge XANES spectra of np-MoS₂ and Ru/np-MoS₂. TFY represents total fluorescence yield mode. **f** SVs characterization by atomic HAADF-STEM image and corresponding intensity line profiles. **g** The confirmed strain of Ru/np-MoS₂ using intensity line profiles at different regions. **h** Atomic HAADF-STEM image of Ru/np-MoS₂, showing the existence of MVs (red circles) and isolated Ru atoms (white circles). **i** Illustration of the spontaneous reduction reaction. Scale bars: (**a**) 500 nm, (**b**) 2 nm, (**c**) 2 nm, (**h**) 1 nm.

spectroscopy elemental analysis, emphasizing the homogeneous distribution of Mo, S, and Ru across the analyzed zone (Supplementary Fig. 12).

The electronic and atomic coordination structure of catalysts were analyzed by XPS, XANES spectroscopy, and extended X-ray absorption fine structure (EXAFS) spectroscopy. The XPS Ru $3p$ peak of Ru/np-MoS₂ shows a ~2.1 eV positive energy shift in relation to that of nanoporous MoS₂ supported Ru nanoparticles catalyst (denoted as Ru_NP/np-MoS₂, Supplementary Fig. 13) (Fig. 3a), indicating the strong interaction between Ru atoms and np-MoS₂, which change the charge density of the Ru atoms. Figure 3b displays the Ru K-edge XANES spectra for Ru/np-MoS₂, RuCl₃, RuO₂, and Ru foil. The absorption-edge of Ru/np-MoS₂ locates between the RuCl₃ and RuO₂, suggesting the oxidation of Ru species after doping in MoS₂. By injecting electrons from Ru species into the MoS₂ substrates, Mo species are reduced and cause phase conversion to form 1T-MoS₂, accompanied with the formation of SVs[23]. The corresponding Fourier transform EXAFS (FT-EXAFS) spectrum for Ru/np-MoS₂ shows a prominent peak at ~1.60 Å, which is attributed to the Ru-S scattering feature, revealing the isolated dispersion of Ru atoms in Ru/np-MoS₂ (Fig. 3c)[34]. The emergence of Ru–Mo

scattering feature is consistent with substitutional doping of Ru into the Mo location. The FT-EXAFS fitting further identifies that four S atoms coordinated with the isolated Ru atoms, while two S atoms loss thus forming the SVs (Supplementary Fig. 14). The Mo K-edge XANES spectra in Fig. 3d show four characteristic peaks with quite different spectral features for Ru/np-MoS₂ and np-MoS₂. Peaks A and D decrease in Ru/np-MoS₂ as compared to np-MoS₂, indicating the increase of 1T-MoS₂ in Ru/np-MoS₂ compared with np-MoS₂[35,36] (Supplementary Fig. 15). This is also evidenced by the XPS results which show the negative energy shifts of Mo $3d$ and S $2p$ peaks of Ru/np-MoS₂ compared with np-MoS₂ (Supplementary Fig. 16)[18]. The broadening of peak B implies the atomic rearrangement after the introduction of Ru atoms[36]. Meanwhile, the rising edge of Ru/np-MoS₂ shifts to lower energy side compared with np-MoS₂, suggesting strong electronic coupling between Ru atoms and MoS₂[20]. As observed in the Mo K-edge FT-EXAFS analyses in Fig. 3e, two peaks at ~1.90 Å and ~2.92 Å are assigned to Mo–S and Mo–Mo scattering feature in np-MoS₂[24]. The FT-EXAFS spectrum for Ru/np-MoS₂ exhibits the much weaker Mo–S peak intensity in comparison with np-MoS₂, resulting from the formation of abundant SVs and the interfacial effect between Ru species and

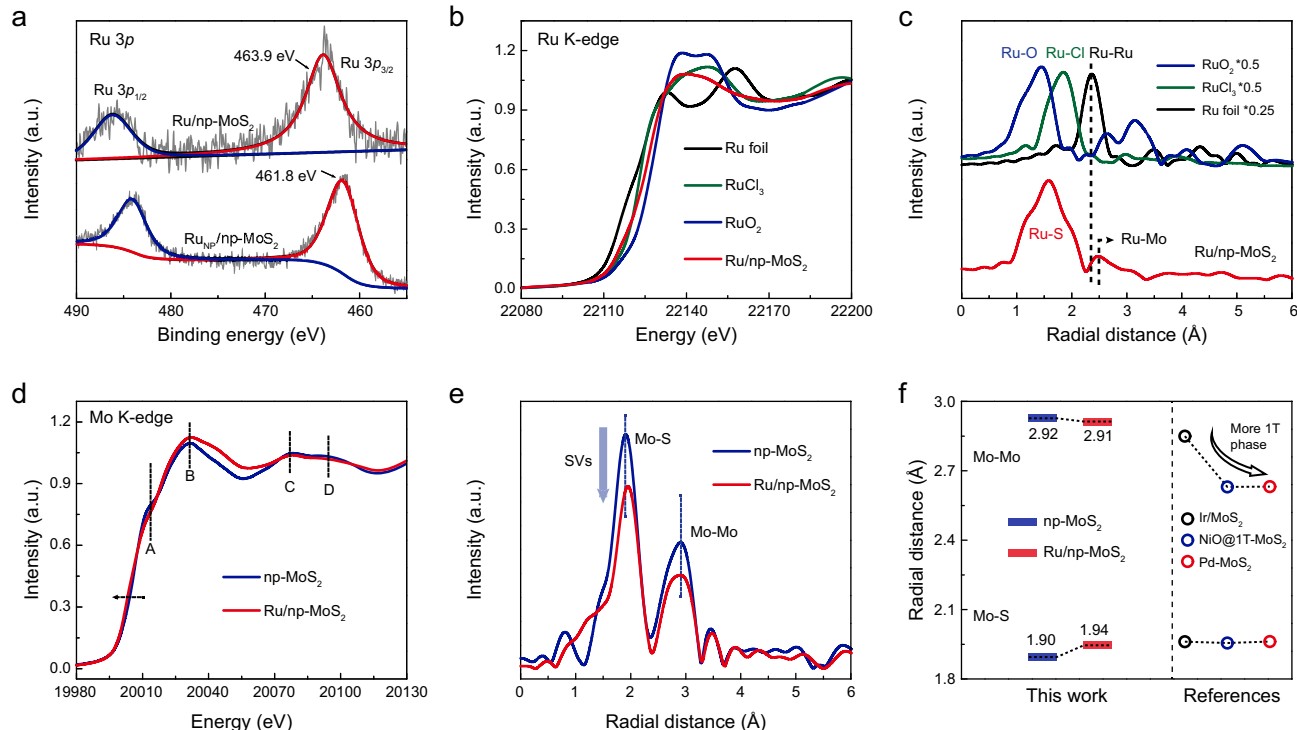

**Fig. 3 X-ray absorption analyses of Ru/np-MoS₂.** **a** Ru 3$p$ XPS spectra of Ru/np-MoS₂ and Ru$_{NP}$/np-MoS₂. **b** Ru K-edge XANES spectra of Ru/np-MoS₂, RuO₂, RuCl₃, and Ru foil. **c** Corresponding FT-EXAFS spectra from (**b**). **d** Mo K-edge XANES spectra of np-MoS₂ and Ru/np-MoS₂. **e** Corresponding FT-EXAFS spectra from (**d**). **f** Comparison of the radial distance of the Mo–S shell and the Mo–Mo shell for np-MoS₂ and Ru/np-MoS₂ with reported MoS₂-based SACs.

MoS₂[20,23]. The decrease in the intensity of the Mo–Mo peak of Ru/np-MoS₂ compared with np-MoS₂ indicates the occurring of structural disorder, which should be attributed to the atomic rearrangement after the substitutional doping of isolated Ru atoms[20]. Besides, the breaking of the Mo–S bond also induces many unsaturated coordinated S, resulting in the high-$R$ shift of Mo–S peak. To precisely display the above process, the variation of Mo–S and Mo–Mo peaks are shown in Fig. 3f. Note that the 2H-1T phase transition would lead to the low-$R$ shift of the Mo–Mo peaks[20,23,24]. Therefore, the slight low-$R$ shift of Mo–Mo peak in Ru/np-MoS₂ as compared with np-MoS₂ is induced by the 2H-1T phase transition.

**Electrochemical performance**. To validate the role of bending strain in boosting the intrinsic activity, control samples of plane MoS₂ (denoted as P-MoS₂) and nanoporous MoS₂ with larger ligament (denoted as Lnp-MoS₂) were prepared, respectively (Supplementary Note 2). Ideally, the strain in P-MoS₂ is negligible, while Lnp-MoS₂ possesses less strain as compared to np-MoS₂. We performed EXAFS spectroscopy to investigate the difference in strain for these support materials. As shown in Fig. 4a, np-MoS₂ exhibits the greatest high-$R$ shift of Mo–Mo peaks among these catalysts. The strain in these catalysts originated from the nanotube-shaped ligament thus formatting the atomically curved MoS₂ (Fig. 4b). The resultant bending strain can be approximately replaced by the tensile strain at the atomic scale (Fig. 4c)[21]. Therefore, the ligament with a smaller diameter ($D_2 < D_1$) possesses the most strained surface atom-arrangement, namely, the most strained SVs. This change can be detected by using the Mo–Mo radial distance as an indicator, as confirmed by the aforementioned FT-EXAFS results (Fig. 4a and Supplementary Table 2). Subsequently, Ru/P-MoS₂ (Ru content: ~9.1 at%) and Ru/Lnp-MoS₂ (Ru content: ~8.3 at%) with the same Ru load to Ru/np-MoS₂ (Ru content: ~8.0 at%) were prepared by using

the spontaneous reduction strategy (Supplementary Figs. 10, 17, 18). These catalysts were then evaluated for HER using the conventional three-electrode configuration in Ar-saturated 1.0 M KOH electrolytes. As unveiled by the linear sweep voltammetry (LSV) presented in Fig. 4d, Ru/np-MoS₂ displays a zero-onset potential, a low overpotential (30 mV) at a current density of 10 mA cm⁻², and a low Tafel slope of 31 mV per decade (mV dec⁻¹) (Fig. 4e), significantly better than that of Ru/P-MoS₂ and Ru/Lnp-MoS₂. The electrochemically effective surface areas (ECSA) normalized LSV curves were performed to highlight the intrinsic activity (Fig. 4f, Supplementary Fig. 19 and Supplementary Note 3). As shown in Fig. 4g, the ECSA-normalized current density of Ru/np-MoS₂ is larger than those of Ru/Lnp-MoS₂ and Ru/P-MoS₂, indicating the high intrinsic activity of the Ru/np-MoS₂. This result indicates that the HER intrinsic activity depends crucially on the strain magnitude, with higher strain inducing the more variation in atomic and electronic structures of Ru sites and SVs. The ability to readily vary the curvature by changing the ligaments of np-MoS₂ offers a convenient way to fine-tune bending strains, which may amplify the synergistic interaction between single-atom Ru and SVs, thus optimizing the HER activity.

Besides, the catalytic performance of Ru/np-MoS₂ far surpasses that of np-MoS₂ and even the commercial catalysts (Ru/C and Pt/C) in terms of overpotential at a current density of 10 mA cm⁻² and Tafel slope. The lower Tafel slope and charge transfer resistance (Supplementary Fig. 20) of Ru/np-MoS₂ as compared to that of np-MoS₂ demonstrate that Ru/np-MoS₂ is endowed with the favorable fast hydrogen evolution kinetics by the introduction of isolated Ru atoms. As reflected in electrochemical double layer capacitance ($C_{dl}$) from the cyclic voltammetry (CV) studies (Supplementary Figs. 19, 21), the Ru/np-MoS₂ is found to possess a larger $C_{dl}$ (15.35 mF cm⁻²) than np-MoS₂ (7.35 mF cm⁻²), indicating more accessible active sites from the Ru atoms and SVs (exposed Mo atoms) in the MoS₂ basal planes.

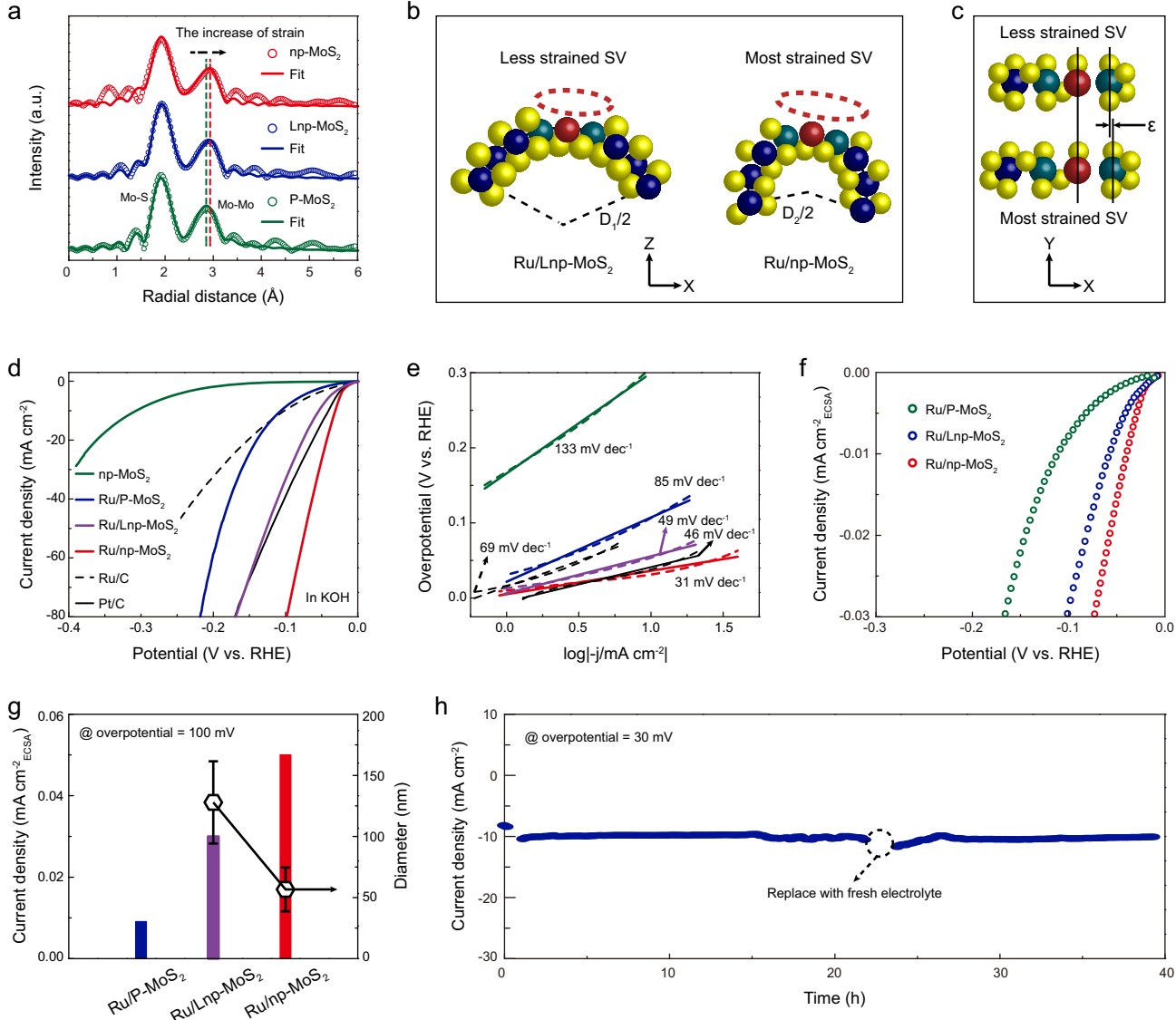

**Fig. 4 Catalytic HER performances. a** The FT-EXAFS spectra of np-MoS$_2$ in comparsion with Lnp-MoS$_2$ and P-MoS$_2$. Corresponding FT-EXAFS fitting curves also shown in Fig. 4a. **b, c** Schematic of the atomic structure of Ru/Lnp-MoS$_2$ and Ru/np-MoS$_2$ derived from (**a**). The ε in (**c**) represents the amount of deformation. **d** Polarization curves of Ru/np-MoS$_2$ as compared with np-MoS$_2$, Ru/P-MoS$_2$, Ru/Lnp-MoS$_2$, Ru/C, and Pt/C. **e** Corresponding Tafel plots derived from (**d**). **f** ECSA-normalized polarization curves of Ru/np-MoS$_2$, Ru/Lnp-MoS$_2$, and Ru/P-MoS$_2$. **g** ECSA-normalized current density at a overpotential of 100 mV for Ru/np-MoS$_2$ in comparsion with those of Ru/P-MoS$_2$ and Ru/Lnp-MoS$_2$. The average diameters of ligaments for Ru/Lnp-MoS$_2$ and Ru/np-MoS$_2$ were also shown in (**g**). Error bars represent the standard deviation from multiple measurements. **h** Current-time response of Ru/np-MoS$_2$ at an overpotential of 30 mV.

In order to probe the impact on performance improvement due to 2H to 1T transition apart from induced strain, we performed the control experiment by comparing the ECSA-normalized current density of nanoporous MoS$_2$ and Ru doped MoS$_2$ under the same strain condition (Lnp-MoS$_2$ and Ru/Lnp-MoS$_2$/np-MoS$_2$ and Ru/np-MoS$_2$) (Supplementary Fig. 22). Obviously, both Ru/Lnp-MoS$_2$ and Ru/np-MoS$_2$ show the increase of current density as compared with Lnp-MoS$_2$ and np-MoS$_2$ due to the formation of local Ru/1T-MoS$_2$ active structure. Meanwhile, it is distinct that Ru/np-MoS$_2$ shows more increment of current density after the formation of local Ru/1T-MoS$_2$ active structure than that of Ru/Lnp-MoS$_2$. The phase transition in Ru doped MoS$_2$ results from the substitutional doping of Ru. By controlling the Ru content of Ru/np-MoS$_2$ and Ru/Lnp-MoS$_2$, the content of 1T-MoS$_2$ in Ru/Lnp-MoS$_2$ is less but very close to that of Ru/np-MoS$_2$ (Supplementary Fig. 23). This indicates that the most

strained Ru/1T-MoS$_2$ active structure in Ru/np-MoS$_2$ displays higher catalytic activity than less strained Ru/1T-MoS$_2$ active structure in Ru/Lnp-MoS$_2$, further highlighting the role of strain in boosting the catalytic activity of active structure. The above merits of Ru/np-MoS$_2$, including overpotential and Tafel slope, are superior to most previously reported MoS$_2$-based catalysts and SACs (Supplementary Table 3). In addition, gas chromatography was introduced to analyze the H$_2$ production, which shows that the H$_2$ Faraday efficiency of Ru/np-MoS$_2$ is close to 100% under different applied potentials (Supplementary Fig. 24). The stability of Ru/np-MoS$_2$ was evaluated by chronoamperometric test, which displays excellent catalytic stability for alkaline HER (Fig. 4h). The XANES and FT-EXAFS spectra of Ru/np-MoS$_2$ after long-time operation shows that the single-atom Ru sites remain atomic dispersion without aggregation (Supplementary Fig. 25), further demonstrating the stability of Ru/np-MoS$_2$.

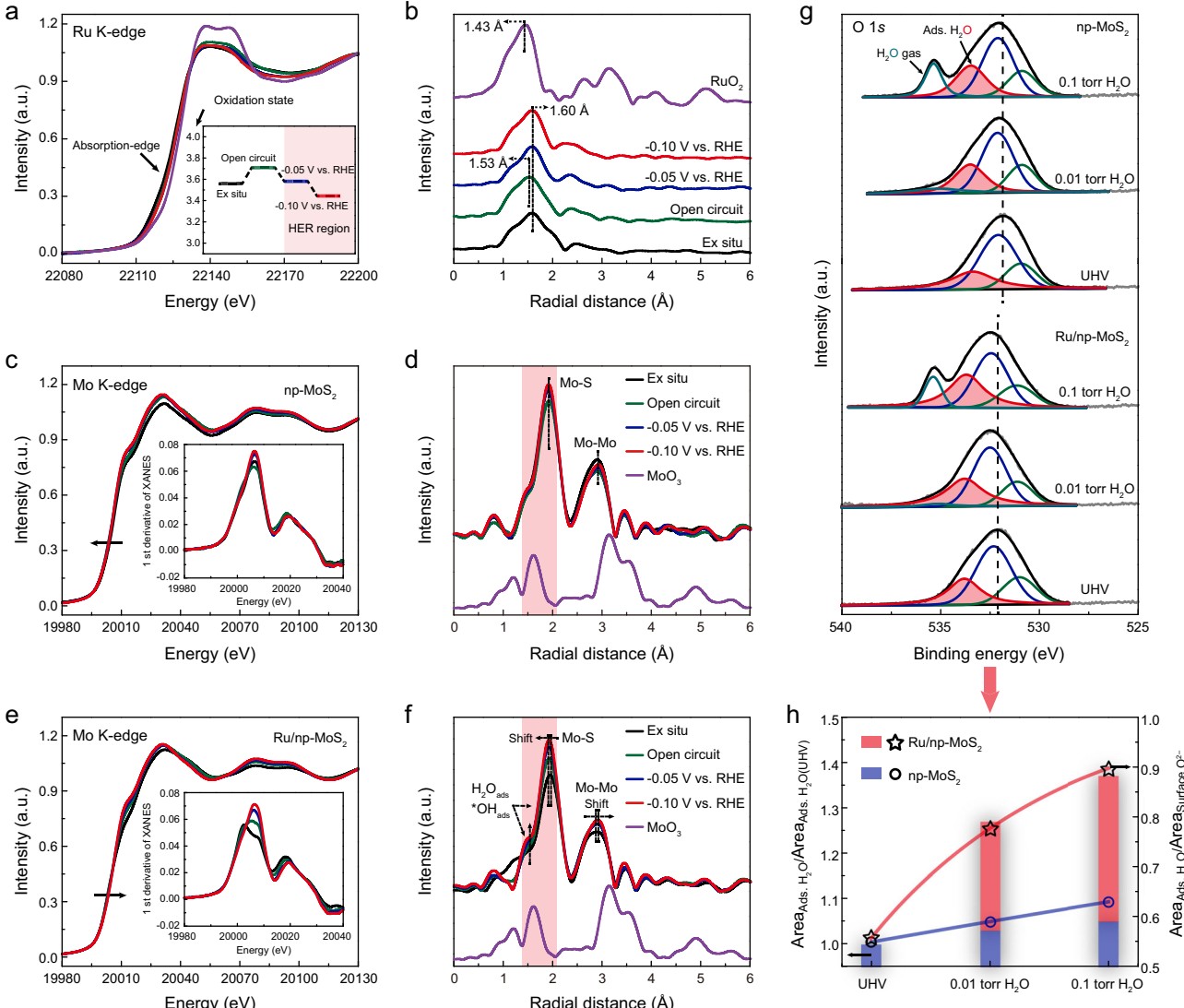

**Fig. 5 Mechanism analyses by operando techniques. a** Operando Ru K-edge XANES spectra of Ru/np-MoS$_2$ recorded at different applied voltages. Inset shows the fitted oxidation states from (**a**). **b** Corresponding FT-EXAFS spectra from (**a**). **c** Operando Mo K-edge XANES spectra of np-MoS$_2$ recorded at different applied voltages. Inset shows the corresponding first-order derivatives of the XANES spectra. **d** Corresponding FT-EXAFS spectra from (**c**). **e** Operando Mo K-edge XANES spectra of Ru/np-MoS$_2$ recorded at different applied voltages. Inset shows the corresponding first-order derivatives of the XANES spectra. **f** Corresponding FT-EXAFS spectra from (**e**). **g** Ambient pressure O 1s XPS experiments of np-MoS$_2$ and Ru/np-MoS$_2$ recorded at different conditions. **h** Comparison of water absorptive capability for np-MoS$_2$ and Ru/np-MoS$_2$.

**Operando XAS tracking of active sites**. To identify the active sites and mechanistically understand the enhancement of HER performance of Ru/np-MoS$_2$, operando XAS measurements were performed by using a home-built cell[6,37]. During the measurements, the operando XAS data were collected under the open circuit condition and two representative potentials (−0.05 and −0.10 V versus reversible hydrogen electrode (RHE)). Figure 5a presents the operando XANES spectra of Ru/np-MoS$_2$ at Ru K-edge, along with commercial RuO$_2$ as reference. Compared with the ex situ condition, the absorption-edge of Ru/np-MoS$_2$ under the open-circuit condition shows a positive-shift, indicating an increase of the Ru oxidation state. This probably results from the binding of H$_2$O and OH$^−$, leading to the delocalization of electron[38–40]. When cathodic potentials of −0.05 and −0.10 V vs. RHE were applied, a negative-shift of the absorption-edge is occurred, indicating the recovery of low-oxidation-state Ru after water dissociation occurred. Note that the catalysts always experience the reduction trend of the cathodic voltage under the

operando HER measurement, which is mainly responsible for the recovery of low-oxidation-state Ru. Under this case, even though there is still H$_2$O adsorption on Ru site, the adsorption of H$_2$O and OH$^−$ cannot balance the reduction trend of the cathodic voltage[40]. To precisely determine the Ru valence state, the fitted oxidation states from the analyses of absorption energy are shown in the inset of Fig. 5a and Supplementary Fig. 26[41]. Corresponding FT-EXAFS spectra for Ru/np-MoS$_2$ at different applied potentials are shown in Fig. 5b. In comparison with the ex situ condition, the main peak obtained under open-circuit condition displays a low-$R$ shift, which is ascribed to the contribution of Ru–O bond (from the binding of H$_2$O and OH$^−$) that overlapped with Ru–S bond. The contribution of Ru–O scattering also leads to the slight increase of the intensity of the main peak[38,40] (Supplementary Fig. 27). During electrochemical H$_2$O reduction (−0.05 and −0.10 V vs. RHE), the peak shows a high-$R$ shift by 0.07 Å. This indicates the distortion of coordination environment for Ru atoms, resulting from the redistribution of the electrons in

Ru atoms between S ligands and the Ru–O bond (from adsorbed $H_2O$ and $OH^-$) under alkaline HER[38,40].

The operando XAS measurements of np-$MoS_2$ and Ru/np-$MoS_2$ at Mo K-edge were conducted to reveal the nature of $MoS_2$ basal planes before and after the introduction of Ru atoms. Figure 5c shows the operando XANES spectra of np-$MoS_2$ at Mo K-edge. There is a negative-shift of rising edge under open-circuit condition compared with that under ex situ condition (Supplementary Fig. 28), indicating the decrease in the Mo oxidation state. It should be noted that the location of Mo sites (central sublayer) hinders the $H_2O$ adsorption and dissociation due to the steric effect in np-$MoS_2$. Thus, the change of Mo oxidation state may result from the interaction between S atoms (outermost sublayer) and electrolyte[42]. When the cathodic potentials (−0.05 and −0.10 V vs. RHE) were applied, the rising edge of np-$MoS_2$ still locates at the lower energy side compared with that under ex situ condition. Correspondingly, the FT-EXAFS spectra of np-$MoS_2$ remain substantially unchanged (Fig. 5d). These suggest that Mo atoms in basal planes (including 2H-$MoS_2$ and 1T-$MoS_2$) of np-$MoS_2$ are inert.

Then, operando XANES spectra of Ru/np-$MoS_2$ at Mo K-edge are presented in Fig. 5e. The rising edge of Ru/np-$MoS_2$ displays a positive-shift under open-circuit condition in relation to that under ex situ condition, meaning an increase of the Mo oxidation state (Supplementary Fig. 28). This is more obviously indicated by the first-order derivatives of the XANES spectra (inset of Fig. 5e). Different from np-$MoS_2$, the Mo sites in Ru/np-$MoS_2$ are exposed due to the formation of SVs. Thus, this change probably results from the binding of $H_2O$ and $OH^-$. When the cathodic potentials were applied (−0.05 and −0.10 V vs. RHE), the rising edge of Ru/np-$MoS_2$ further shifts to higher energy. This implies the further increase of the Mo oxidation state in Ru/np-$MoS_2$ during the HER. Combining the DFT prediction, we think that there are abundant $H_2O$ and $OH^-$ adsorbed on Mo sites without the subsequent dissociation, which balances the reduction trend of the cathodic voltage resulting in the further increase of oxidation state[40]. The FT-EXAFS spectra of Ru/np-$MoS_2$ were presented in Fig. 5f. Under open-circuit condition, the rise of peak for $H_2O$ and $OH^-$ adsorption is detected, further supporting the above conclusions. When the cathodic voltage was applied (−0.05 V vs. RHE) and subsequently increased (−0.10 V vs. RHE), the peak intensity for $H_2O$ and $OH^-$ adsorption further increased, meaning the enhancing adsorption of $H_2O$ and $OH^-$. During this process, the O atoms from $H_2O$ or $OH^-$ occupied the location of strained SVs. Meanwhile, the negative-shift of Mo–S peak and the positive-shift of Mo–Mo peak occurred. This suggests that much $H_2O$ or $OH^-$ enter in strained SVs in $MoS_2$ basal planes and bind with Mo atoms, leading to the distorted coordination environment of Mo atoms.

To further verify the results, ambient pressure XPS (AP-XPS) was performed to investigate the adsorption of $H_2O$ (Supplementary Figs. 29, 30)[43]. Figure 5g displays the curve-fitting of the O 1s XPS spectra for np-$MoS_2$ and Ru/np-$MoS_2$ under ultrahigh vacuum (UHV), a water pressure of 0.01 torr, and a water pressure of 0.1 torr. The increased chemisorbed $H_2O$ on the surface of both np-$MoS_2$ and Ru/np-$MoS_2$ with the increase of water pressure is observed, manifesting as the increase of the area of adsorbed $H_2O$ and the shift of main peak toward high energy in the O 1s XPS spectra. Furthermore, the area of the corresponding peak in the spectra is utilized to precisely determine the increase of adsorbed $H_2O$ by comparing the area of adsorbed $H_2O$ under different conditions and the area of adsorbed $H_2O$ under UHV (or the area of adsorbed $H_2O$ and the area of surface $O^{2-}$ at same conditions) (Fig. 5h). It is clear that Ru/np-$MoS_2$ could adsorb more $H_2O$ than np-$MoS_2$ under the same conditions, further confirming that the enhancement of

water adsorption is responsible for the superior catalytic activity of Ru/np-$MoS_2$.

Our operando XAS and AP-XPS results indicate that Ru sites and the Mo sites located below the SVs are the active sites for alkaline HER. The formation of SVs around Ru atoms in Ru/np-$MoS_2$ plays a vital role in the $H_2O$ dissociation processes on Ru atoms. Because the effective mass transfer of $H_2O$ molecules and $OH^-$ groups to the active Ru atom is a key factor that determines alkaline hydrogen evolution activity[34,44]. The remarkable enrichment of $H_2O$ in SVs around Ru atom could improve the water mass transfer for the subsequent alkaline HER, manifesting as the easier binding between Ru atoms and $H_2O$. These results further support the aforementioned theoretical analyses. Significantly, we demonstrate how the strain affects the synergetic interaction between single-atom Ru sites and SVs by combining theoretical analyses with in situ techniques.

## Discussion

In this work, a strain engineering strategy was developed to amplify the synergetic effect between single-atom Ru sites and SVs based on a nanoporous $MoS_2$-based Ru SAC system. The bending strain induced from curved ligaments of nanoporous $MoS_2$ can modulate the interaction between single-atom Ru sites and SVs, thus enhancing catalytic activity of catalyst. The best catalyst, Ru/np-$MoS_2$, delivers an overpotential of 30 mV to achieve a current density of 10 mA $cm^{-2}$ and a Tafel slope of 31 mV $dec^{-1}$ for alkaline HER, surpassing the state-of-the-art catalyst and commercial catalyst. By virtue of theoretical analyses, electrochemical experiments, and in situ techniques, we identified that the synergetic effect between Ru sites and SVs manifested as the water mass transfer from SVs to Ru sites and subsequent water dissociation process on Ru sites. The bending strain accelerates water mass transfer and water dissociation at the same time, thus achieving the amplifying of synergetic effect. On the basis of these design principles, this strategy can be explored by altering the assisting sites or the modulating method, thus achieving the maximum activity of SACs.

## Methods

**Materials syntheses**. The chemically dealloyed 3D nanoporous gold (NPG) was used as substrates for the chemical vapor deposition of monolayer $MoS_2$[31,45]. Then, the free-standing monolayer $MoS_2$@NPG composites were etched by $I_2$-KI solution (12 mg $I_2$ and 6 mg KI dissolved in 100 mL deionized water) for 24 h to obtain monolayer $MoS_2$ with nanoporous structure (np-$MoS_2$) (Supplementary Fig. 6). Finally, the Ru/np-$MoS_2$ was synthesized through a spontaneous reduction method. In brief, $RuCl_3·H_2O$ (3 mg) was placed in a flask with 50 mL deionized water and stirred for 2 h. Afterward, np-$MoS_2$ film was transferred to the above solution to adsorb Ru species at room temperature for 12 h. The as-obtained film was transferred to the carbon cloth and dried for 12 h under room temperature and atmospheric pressure. Finally, the sample was dried for 12 h under vacuum in an oven at 60 °C to produce Ru/np-$MoS_2$. As a comparison, $Ru_{NP}$/np-$MoS_2$ was synthesized through an electrochemical deposition method. In a typical synthesis, the Ru loading on np-$MoS_2$ was carried out with a three-electrode system using an electrochemical workstation (Ivium CompactStat. h). The np-$MoS_2$ was coated on the carbon cloth to form a working electrode. An Ag/AgCl electrode and a carbon rod were used as reference and counter electrodes, respectively. For the Ru source, 5 mg of $RuCl_3$ was poured into 200 mL of a 0.5 M $H_2SO_4$ electrolyte. The electrochemical deposition process was carried out by 100 CV cycles with a voltage range from 0.0 to −0.6 V vs. RHE at a scan rate of 50 mV $s^{-1}$.

**Characterizations**. SEM measurements were performed on a Zeiss Sigma HD SEM. HAADF-STEM images and EDX mapping were taken by a JEM-ARM 200 F with double spherical aberration correctors. XPS measurements were performed on Thermo Scientific ESCALAB250Xi spectrometer equipped with an Al Kα monochromatic.

**Electrochemical measurements**. The HER activity and durability were measured using an electrochemical workstation (Ivium CompactStat. h). The catalysts were coated on the carbon cloth to form a working electrode. A saturated calomel electrode (SCE) and a carbon rod were used as reference and counter electrodes,

respectively. LSV curves were obtained in 1.0 M KOH solutions at a scan rate of 2 mV s$^{-1}$, de-aerated with Ar at room temperature.

**Operando XAS measurements**. The Ru K- and Mo K-edges XAS spectra were measured at the beamline BL01C1 of National Synchrotron Radiation Research Center (NSRRC, Taiwan). The Mo L$_3$- and S K-edges XAS spectra were measured at the beamline BL16A1 of NSRRC. Operando XAS measurements were performed using an electrochemical workstation (Ivium CompactStat. h) and a home-built cell[6,37]. An SCE and a carbon rod were used as reference and counter electrodes, respectively. The working electrode was prepared by coating the catalyst on the carbon cloth. The fresh 1.0 M KOH electrolyte was bubbled with Ar for 1 h. For the installation of operando XAS setup, the side of the working electrode covered with Kapton film was faced to the incident X-rays, while the other side of the working electrode was contacted with electrolyte. The XAS spectra were measured in the fluorescence mode at room temperature. During the operando experiments, the different potentials of open circuit, −0.05, and −0.10 V vs. RHE were applied to the system. Acquired XAS data were processed with the ATHENA program.

**AP-XPS measurements**. AP-XPS measurements were performed at the 24A1 beamline of NSRRC (Supplementary Fig. 29). The Ru/np-MoS$_2$ and np-MoS$_2$ film were directly covered on the carbon conductive adhesive, thus avoiding the influence of carbon conductive adhesive signal. Then, the sample holder loaded with Ru/np-MoS$_2$, np-MoS$_2$, and Au foil was exposed in the analysis chamber. The AP-XPS analyses were conducted under UHV, a water pressure of 0.01 torr, and a water pressure of 0.1 torr, respectively. The obtained XPS data were corrected by using the Au 4f XPS spectrum of an Au foil.

## Data availability
The data that support the findings of this study are available from the corresponding authors upon reasonable request.

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

## Acknowledgements
The authors gratefully acknowledge financial support by the National Natural Science Foundation of China (Grant No. 51771072), the Youth 1000 Talent Program of China,

the Outstanding Youth Scientist Foundation of Hunan Province (Grant No. 2020JJ2006), the Fundamental Research Funds for the Central Universities, and Hunan University State Key Laboratory of Advanced Design and Manufacturing for Vehicle Body Independent Research Project (Grant No. 71860007). The authors thank Dr. Chia-Hsin Wang for the AP-XPS experiments at Taiwan Light Source.

## Author contributions

Y.W.T. conceived and supervised this study. K.J. and D.C.C. carried out materials fabrication. K.J. performed SEM/TEM/XPS characterizations and electrochemical measurements. M.L. and Z.L. performed the DFT calculations. K.J., M.P., Y.R.L., T.S.C., and F.M.F.de G. contributed to the XAS measurements and analyses of the XAS experiments results. K.J. performed the AP-XPS experiments and analyses. Y.W.T. and K.J. wrote the paper. All authors contributed to discussions and manuscript review.

## Competing interests

The authors declare no competing interests.
