## [Peer Review File · Nature Communications]

REVIEWER COMMENTS

Reviewer #1 (Remarks to the Author):

The manuscript entitled "Rational strain engineering of single-atom ruthenium on nanoporous MoS₂ for highly efficient hydrogen evolution" presents a class of ruthenium single atom catalysts on nanoporous MoS₂ for the alkaline hydrogen evolution reaction. The goal of this study is to demonstrate how sulphur vacancies (SVs) and structural strains affect the performance of these materials by using synchrotron-based techniques (XAS, XPS), transmission electron microscopy techniques, and DFT calculations. It is an interesting topic, but a number of issues should be addressed:

-Line 136: L3 spectra of reference 23 are clearly different. In this case the decrease of the white line of Mo L3-edge and S K-edge seems to be due to a not good normalization.

-Lines 138-140: the loss of S atoms after Ru doping revealed by HAADF-STEM does not demonstrate that SVs are around Ru

-Lines 156-157: any idea of Ru nanoparticles size in RuNP/np-MoS?

-Why, together with RuCl₃, RuO₂, and Ru metal foil as XAS references, RuS₂ has not been used?

-Lines 164-167: errors on the coordination numbers (Supplementary Fig. 13) can be high and should be indicated (but also errors on the other parameters). It is difficult to assign without any doubt the small peak around 2.5 Angstrom in the FT-EXAFS to Ru-Mo distance. A fit is necessary, also to validate the model proposed in Sup Figure 13

-Lines 169-171: it is difficult to appreciate the spectral differences in points C and D (Fig 3d). What about the peak B? This peak behaves differently from that of reference 35, and in reference 35 the S K-edges (not shown in this study) are more convincing in revealing the 2H-1T transition.

-From line 181 it is discussed that the FT-EXAFS signal shows a weaker Mo-Mo and Mo-S peak intensity in comparison with np-MoS₂. This difference is attributed to SVs formation and 2H-1T phase transition. This decrease in peaks intensity is strong, and cannot be due to SVs formation only; indeed, how the Mo-Mo signal decreases can be linked to SVs formation? 2H and 1T phases have the same coordination number, so why this decrease should be indicative of such transition?

-From line 195 EXAFS is used to investigate strain effects. Differences in Figure 4 are not, in my opinion, incisive: small and just qualitative changes, so that a fit (also showing the errors) should be performed.

-Line 215: "demonstrating the role of bending strain in boosting the intrinsic activity of the catalyst". This is not fully demonstrated, because the activity might also depend on defects, structural changes, amorphization.

-Line 239: "The XANES and FT-EXAFS spectra of Ru/np-MoS₂ before and after long-time operation remain substantially unchanged (Supplementary Fig. 20)". This is not true, and spectra should be superposed for better comparison. The second peak of the FT at Ru edge around 2.5 Angstrom is shifted (and probably also the first shell peak at Mo K-edge). This peak was assigned to Ru-Mo distance, and this behavior should be explained.

-As far as the operando XAS measurements is concerned: 1) why measurements were carried out at -.05 and -.1 V only? Why not at lower or other intermediate potentials? 2) a probably binding of H₂O and OH is not supported by any changes in the coordination number of FT-EXAFS Ru K-edge? 3) from line 279: "the rising edge of Ru/np-MoS₂ displays a positive-shift[...] due to the formation of strained SVs, leading to direct binding between H₂O and Mo atoms". The shift is difficult to detect, and, if real, it indicates a change in the oxidation state, which is not the direct evidence of a binding between Mo and H₂O. 4) Line 287: "Under open-circuit condition, the rise of peak for H₂O and OH- adsorption is detected, indicating that the exposed Mo atoms act as active sites for H₂O adsorption". Increase in the coordination number or FT intensity does not prove that this is the active site. The proposed mechanism, during which H₂O or OH enter in strained SVs of MoS₂ plane, cannot be only proved by shifts in the FT-EXAFS signals, because structural changes may be just induced by the applied potential.

-It would be very useful if AP-XPS had been carried out under applied potential.

-A lack of this study is the absence of experiments performed at the S K-edge.

Reviewer #2 (Remarks to the Author):

In this work, the authors have developed a strain engineering strategy to investigate the synergetic effect between single-atom Ru sites and SVs based on the Ru/np-MoS₂ sample. The successful introduction of strain is carefully demonstrated by adequate structural and spectroscopy characterizations. To strengthen the understanding on the enhancement mechanism, the DFT calculations are combined with the operando XAS and ambient pressure XPS spectra to reveal the possible reaction process. Generally, the results are interesting and the findings here are attractive. However, there are still some scientific inconsistencies in the manuscript, which should be addressed before being further considered to publish in Nature Communication. Please find the detailed comments below.

1. The explanations for the DFT calculation in Figure 1 are not making sense currently.

" It is good that the Ru/np-MoS₂ has such a low Tafel slope of 31 mV dec⁻¹ in Figure 4e. Clearly, it will go through a Volmer-Tafel process rather than a Volmer-Heyrovsky process during HER, suggesting that two adsorbed H at the surface generated from water dissociation are coupled together to form H₂. However, it will never happen if the single Ru site is considered to complete the process alone. Thus, the current explanations on the DFT results is not consistent with the experimental results.

" If you look at the binding energy of H for S site (especially after applying the strain) in Figure 1d, it is a pretty good active site while it is ignored throughout the manuscript. If the Volmer-Tafel process is happening, it is highly possible that the H adsorbed at Ru are coupled with the H adsorbed at nearby S.

" The author have interpreted that the adsorbed H₂O in exposed Mo will dissociate at the Ru site. However, considering the abundance of H₂O in the electrolyte, as well as the more favorable adsorption energy on Ru site, such a prediction is unconvincing and confusing.

" What kind of MoS₂ are used for calculation, 1T or 2H?

2. In Figure 5c and 5e, the trend in the rising edge is not clear, please consider modify the layout of these figures. The information delivered from the insets is also ignored. The current explanations on these operando XANES data are a little awkward:

" If the Mo is really inert in np-MoS₂, it needs to explain the shift in the data and why it is reduced.

" If the S is the site to bind with H₂O, it is unfair to be deduce from the results of Mo K-edge XANES.

It has already been predicted that S sites are has more favorable binding energy for water dissociation and H-H coupling (Figure 1d). However, the important operando XAS for S edge are missing.

" Hypothetically, if the signals of Mo K-edge XAS spectra are true for Ru/np-MoS₂ in Figure 5e-f, it might not be a good sign. The obvious change due to the O species binding at the surface suggests the poisoning of the exposed Mo.

" Please make sure the data processing for Figure 4a, Figure 5d and 5f are the same and avoid the over-interpretation of these data.

3. About the oxidation state of Ru.

" The peak of the Ru/np-MoS₂ is shifted to higher binding energy for Ru 3p XPS spectrum in Figure 3a, while those for Mo 3d and S 2p in Supplementary Figure 14 are shifted oppositely. Thus an electron injection should be responsible for the formation of 1T phase. Unfortunately, this issue is not included in the discussion. In this case, the Ru are losing electron and being oxidized?

" In the inset of Figure 5a, please provide more details about how to get the oxidation state of Ru (> 3+). Clearly, since the precursor to prepare Ru SACs is RuCl₃, the Ru are oxidized. It would be contradictory to the formation mechanism for Ru SACs and SVs (the reduction of Ru?). Please take good care of these judgements to avoid misunderstandings.

4. In Figure 3e, where does the 2H-MoS₂ come from? The comparison between the as-prepared np-MoS₂ samples and 2H-MoS₂ is not fair enough to illustrate the change in local bonding are due to the strain. If so, it need to explain why there are contraction in Mo-S bond but extension in Mo-Mo bond.

5. Surprisingly, the Ru content is actually very high (~ 8 at. %) for single-atom catalysts. Please

make a more precisely comparison among literature and explain what happens in the as-prepared samples.

6. Just for curiosity, how about the HER performance in acid? The authors try so hard to persuade that the strain can enhance the water adsorption properties of SVs and exposed Mo, it would be worthy to give it a shot at the acidic media, which might be helpful to support the claims that were made in this manuscript.

7. How to tell the impact on performance improvement due to 2H to 1T transition apart from induced strain?

8. The author have made great efforts to demonstrate the existence and importance of strain. How about the impact of specific strain (such as pressure stress and tensile stress). What kind of stress are taking about in this manuscript? How about the explorations of these stress in the literature?

Reviewer #3 (Remarks to the Author):

This is a joint theory and experimental work on single atom catalysis, Ru doped in MoS₂. Finally, the importance of this work is to be judged based on what has been achieved experimentally. My comments, however, will be on the computational aspects.

In my opinion, several crucial technical details about the DFT calculations have been left out, which hampers a full understanding, and reproduction if someone is interested, of the results and their validity. I am listing these below.

The authors claim that they 'hypothesized that the introduction of isolated Ru atoms into MoS₂ could cause the loss of S atoms around Ru atoms'. It is not at all clear why such a hypothesis is physically reasonable. (Though they have shown subsequently that the experiments suggest so). Accepting that it is a reasonable hypothesis, an a posteriori validation could have been provided from DFT calculations. The authors have not done that. Or else, they can simply write that such a scenario was suggested by experimental observations. Since they have placed the theory first, making it the guide for subsequent experiments, a theoretical justification for such a hypothesis has to be presented.

On the same page they write 'Ru sites of Ru/MoS₂ possess much lower water adsorption energy of -0.516 eV as compared to that of MoSV sites ...' and refer to Fig. 1(b) and Supplementary Fig1. Now Fig. 1 only shows the structure, gives no quantitative information about water adsorption. Supplementary Fig. 1 shows calculated water adsorption energies. The final argument about efficacy of Ru sites over Mosv in H₂O adsorption, dissociation and subsequent HER is based on the free energies presented in Fig 1(d), which is the correct way of looking at it. How are the free energies derived from the energies presented in Suppl. Fig 1? It is important to tell the reader how the zero point energy and entropy contributions are calculated or obtained.

Neither Fig 1(d) nor Suppl. Fig 1 tells the amount or the nature of the strain.

The authors write 'Even if the energy barrier of water dissociation for MoSV site slightly decreases after the applied strain, its value still ...' The free energies reported in Fig 1(d), by themselves, do not give the barriers for H₂O dissociation or the HER process, in my opinion. These are the free energies of some intermediate steps in the whole process. There may be (generally are) further kinetic barriers in between. The authors must clarify this, re-write this part so as not to overstate the results.

They also write within the section on theoretical results 'The subsequent H-H coupling step also highlights the role of strain ...' How? I do not see this addressed within DFT.

Details about what system they have taken for DFT calculations have not been reported. These are customary. Is it a single layer of MoS₂ or a nanotube? What is the size? What is the size of the

vacuum layer in the non-periodic direction? What are the formation energies of Ru substitution, S vacancy etc.? Such details must be reported in the Suppl. Info.

As I said, the main validation of the work has to come from the experimental part. But, in addition, the above shortcomings in the computational part have to be addressed before it becomes suitable for publication.

Responses to the Referees' Comments

We thank the referees for their valuable comments and positive endorsement to our manuscript. We have carefully considered the referees' comments and revised the manuscript accordingly. Our responses and corresponding revisions are as follows:

Reviewer #1:

The manuscript entitled "Rational strain engineering of single-atom ruthenium on nanoporous MoS₂ for highly efficient hydrogen evolution" presents a class of ruthenium single atom catalysts on nanoporous MoS₂ for the alkaline hydrogen evolution reaction. The goal of this study is to demonstrate how sulphur vacancies (SVs) and structural strains affect the performance of these materials by using synchrotron-based techniques (XAS, XPS), transmission electron microscopy techniques, and DFT calculations. It is an interesting topic, but a number of issues should be addressed:

Response: Thank you for your positive comments on our manuscript. We have revised our manuscript accordingly.

Comment 1. Line 136: L₃ spectra of reference 23 are clearly different. In this case the decrease of the white line of Mo L₃-edge and S K-edge seems to be due to a not good normalization.

Response: Thank you for your comments. After carefully checked our data, we have revised the S K- and Mo L₃-edges XANES spectra in the revised manuscript. For your convenience, we also show the spectra as following:

Figure R1. XAS characterizations of np-MoS₂ and Ru/np-MoS₂.

(a) S K- and (b) Mo L₃-edge XANES spectra of np-MoS₂ and Ru/np-MoS₂.

Besides, the Mo L₃-edge spectra of Ref. 23 compared the difference between 2H-MoS₂ and Pd-MoS₂, which shows the obvious phase transition. However, the pristine np-MoS₂ consist of 2H-MoS₂ and 1T-MoS₂ in this work. The introduction of Ru atoms leads to the increase of 1T-MoS₂ in Ru/np-MoS₂ than that of np-MoS₂. Therefore, the variations of Mo L₃-edge spectra in this work are smaller than that of Ref. 23.

Comment 2. Lines 138-140: the loss of S atoms after Ru doping revealed by HAADF-STEM does not demonstrate that SVs are around Ru.

Response: We sincerely appreciate your valuable comment. According to the comment, we have carefully revised our manuscript and the details are listed below:

The magnified HAADF-STEM image of Ru/np-MoS₂ and corresponding intensity profile analyses reveal the loss of S atoms after the substitutional doping of Ru, thus forming SVs (**Fig. 2f**).

The HAADF-STEM only demonstrate the loss of S atoms locally. The formation of SVs around Ru was testified by the FT-EXAFS spectrum at Ru K-edge and corresponding fitting data (**Supplementary Fig. 14**).

Comment 3. Lines 156-157: any idea of Ru nanoparticles size in Ru_{NP}/np-MoS₂?

Response: We appreciate the question from the reviewer. We have added corresponding particle size distribution of Ru nanoparticles in **Supplementary Fig. 13 (Fig. R2)**.

Figure R2. HAADF-STEM characterizations of Ru_{NP}/np-MoS₂.

HAADF-STEM image of Ru_{NP}/np-MoS₂ (a), showing that Ru nanoparticles are uniformly distributed on the ligament of np-MoS₂ with an average diameter of ~6.25 nm (b). Magnified HAADF-STEM image of Ru_{NP}/np-MoS₂ (c), showing many Ru nanoparticles and clusters on the surface of MoS₂. Scale bars: (a) 20 nm, (b) 10 nm.

Comment 4. Why, together with RuCl₃, RuO₂, and Ru metal foil as XAS references, RuS₂ has not been used?

Response: Thanks for your suggestion. We agree that using as a RuS₂ reference is helpful for us to determine the coordination environment of isolated Ru atom. But our group cannot conduct the XAS test in National Synchrotron Radiation Research Center (NSRRC) in Taiwan recently due to the COVID-19. Many thanks for your understanding. Even without RuS₂ as a reference, we believe that the coordination environment of Ru can be determined. We think that the emerge of Ru-Mo peak in the Ru K-edge EXAFS spectrum of Ru/np-MoS₂ is very important. The Ru coordination environment are in good agreement with that of 1T-MoS₂ (Ref. 36). This proved the substitutional doping of Ru into the Mo location. We have conducted the EXAFS spectrum fitting to further identify the local structure of Ru atom (Please see **Comment 5**).

Ref. 36: Li, H. et al. Systematic design of superaerophobic nanotube-array electrode comprised of transition-metal sulfides for overall water splitting. *Nat. Commun.* **9**, 2452 (2018).

Comment 5. Lines 164-167: errors on the coordination numbers (Supplementary Fig. 13) can be high and should be indicated (but also errors on the other parameters). It is difficult to assign without any doubt the small peak around 2.5 Angstrom in the FT-EXAFS to Ru-Mo distance. A fit is necessary, also to validate the model proposed in Sup Figure 13.

Response: We appreciate the suggestions from the reviewer. According to your comments, we revised our FT-EXAFS fitting curves of Ru/np-MoS₂ as following:

Figure R3. FT-EXAFS fitting curves of Ru/np-MoS₂.

The local atomic structure of Ru in Ru/np-MoS₂ derived by EXAFS fitting matches well with the Ru-S₄ model, suggesting the loss of two S atoms around Ru atom thus forming the SVs. The emerge of Ru-Mo bond with a bond length of 2.85 Å in Ru/np-MoS₂ indicates the substitutional doping of Ru atom in 1T-MoS₂ (Ref. 3).

Note: R represents the interatomic distance; CN represents the coordination number; σ^2 represents the Debye-Waller factor; ΔE_0 represents the edge-energy shift.

Ref. 3 in **Supplementary Information**: Li, H. et al. Systematic design of superaerophobic nanotube-array electrode comprised of transition-metal sulfides for overall water splitting. *Nat. Commun.* **9**, 2452 (2018).

Comment 6. Lines 169-171: it is difficult to appreciate the spectral differences in points C and D (Fig 3d). What about the peak B? This peak behaves differently from that of reference 35, and in reference 35 the S K-edges (not shown in this study) are more convincing in revealing the 2H-1T transition.

Response: Thank you very much for this comment. In order to show this spectral feature more clearly, we added the magnified spectra in the revised **Supplementary Information (Fig. R4 / Supplementary Fig. 15)**:

The spectra show that the peak D of Ru/np-MoS₂ decrease to nearly disappearance compared with that of np-MoS₂. This suggests the increase of 1T phase in Ru/np-MoS₂ compared with np-MoS₂ (Ref. 35).

Figure R4. Spectra features of Mo K-edge XANES spectra.

Spectra features of Mo K-edge XANES spectra of np-MoS₂ and Ru/np-MoS₂.

About the peak B, we think that the broadening of peak B implies the atomic rearrangement after the introduction of Ru atoms, which has been reported in previous literature (Ref. 36).

We agree that the S K-edge XANES spectra are more convincing in revealing the 2H-1T transition. Therefore, we added the S K-edge XANES spectra of np-MoS₂ and Ru/np-MoS₂ in **Fig. 2e** to further explain the phase transition in Ru/np-MoS₂.

Ref. 36: Li, H. et al. Systematic design of superaerophobic nanotube-array electrode comprised of transition-metal sulfides for overall water splitting. *Nat. Commun.* **9**, 2452 (2018).

Comment 7. From line 181 it is discussed that the FT-EXAFS signal shows a weaker Mo-Mo and Mo-S peak intensity in comparison with np-MoS₂. This difference is attributed to SVs formation and 2H-1T phase transition. This decrease in peaks intensity is strong, and cannot be due to SVs formation only; indeed, how the Mo-Mo signal decreases can be linked to SVs formation? 2H and 1T phases have the same coordination number, so why this decrease should be indicative of such transition?

Response: Many thanks for these comments. According to these comments, we carefully read relevant literatures and revised these sentences as following:

The FT-EXAFS spectrum for Ru/np-MoS₂ exhibits the much weaker Mo-S peak intensity in comparison with np-MoS₂, resulting from the formation of abundant SVs and the interfacial effect between Ru species and MoS₂ (Ref. 20, 23). The decrease in the intensity of the Mo-Mo peak of Ru/np-MoS₂ compared with np-MoS₂ indicates

the occurring of structural disorder, which should be attributed to the atomic rearrangement after the substitutional doping of isolated Ru atoms (Ref. 20).

Ref. 20: Wei, S. et al. Iridium-triggered phase transition of MoS₂ nanosheets boosts overall water splitting in alkaline media. *ACS Energy Lett.* **4**, 368-374, (2018).

Ref. 23: Luo, Z. et al. Chemically activating MoS₂ via spontaneous atomic palladium interfacial doping towards efficient hydrogen evolution. *Nat. Commun.* **9**, 2120 (2018).

Thank you again for these comments.

Comment 8. From line 195 EXAFS is used to investigate strain effects. Differences in Figure 4 are not, in my opinion, incisive: small and just qualitative changes, so that a fit (also showing the errors) should be performed.

Response: We appreciate the reviewer for this nice suggestion. According to this comment, we performed the FT-EXAFS fitting to further explain the structure of catalysts. The fitting results are shown as following:

Fig. R5. FTEXAFS fitting curves.

The FT-EXAFS spectra of np-MoS₂ in comparison with Lnp-MoS₂ and P-MoS₂. Corresponding FTEXAFS fitting curves also shown in Fig. R5.

Table R1. Structural parameters extracted from the Mo K-edge EXAFS fitting.

Catalysts	Scattering pair	CN	R (Å)	σ^2 (10^{-3} Å ²)	ΔE_0 (eV)	R -factor
P-MoS ₂	Mo-S	5.1±0.7	2.400±0.01	1.55±0.8	0.272	0.005
	Mo-Mo	3.9±0.5	3.149±0.01	2.48±0.8	-2.14	
Lnp-MoS ₂	Mo-S	5.9±0.5	2.405±0.01	2.63±0.6	2.53	0.008
	Mo-Mo	4.1±0.4	3.161±0.01	3.70±0.6	2.43	
np-MoS ₂	Mo-S	5.7±0.7	2.408±0.01	2.66±1.3	2.77	0.010
	Mo-Mo	4.1±0.5	3.169±0.01	3.42±1.3	2.90	

CN represents the coordination number; R represents the interatomic distance; σ^2 represents the Debye-Waller factor; ΔE_0 represents the edge-energy shift.

Comment 9. Line 215: "demonstrating the role of bending strain in boosting the intrinsic activity of the catalyst". This is not fully demonstrated, because the activity might also depend on defects, structural changes, amorphization.

Response: We appreciate the reviewer for this reminding. We are sorry for the loose of this description. We have revised this sentence as following:

The ECSA-normalized current density of Ru/np-MoS₂ is larger than those of Ru/Lnp-MoS₂ and Ru/P-MoS₂, indicating the high intrinsic activity of the Ru/np-MoS₂.

Comment 10. Line 239: "The XANES and FT-EXAFS spectra of Ru/np-MoS₂ before and after long-time operation remain substantially unchanged (Supplementary Fig. 20)". This is not true, and spectra should be superposed for better comparison. The second peak of the FT at Ru edge around 2.5 Angstrom is shifted (and probably also the first shell peak at Mo K-edge). This peak was assigned to Ru-Mo distance, and this behavior should be explained.

Response: Many thanks for this comment. According to this comment, we superposed spectra for comparison. Besides, we also revised this sentence as

following:

The XANES and FT-EXAFS spectra of Ru/np-MoS₂ after long-time operation shows that the single-atom Ru sites remain atomic dispersion without aggregation (Supplementary Fig. 25 / Fig. R6). Further discussions were added below the Fig. R6.

Fig. R6. XAS characterizations after long-time operation.

(a, c) XANES spectra at Ru K- and Mo K-edges. (b, d) Corresponding FT-EXAFS spectra at Ru K- and Mo K-edges.

The FT-EXAFS spectra at Ru K-edges shows the negative-shift of Ru-Mo peak after long-time operation, indicating the shrinkage of interatomic distance between Ru atoms and Mo atoms. This irreversible shrinkage structure may be able to stabilize the isolated Ru atoms. The operation for a long-time also lead to the slight changes in oxidation state and structure of np-MoS₂, manifesting as the positive-shift of rising edge in XANES spectra at Mo-K edge and the low-R shift of Mo-S peak in corresponding FT-EXAFS spectra.

Comment 11. As far as the operando XAS measurements is concerned: 1) why measurements were carried out at -0.05 and -0.10 V only? Why not at lower or other

intermediate potentials? 2) a probably binding of H₂O and OH is not supported by any changes in the coordination number of FT-EXAFS Ru K-edge? 3) from line 279: "the rising edge of Ru/np-MoS₂ displays a positive-shift[...] due to the formation of strained SVs, leading to direct binding between H₂O and Mo atoms". The shift is difficult to detect, and, if real, it indicates a change in the oxidation state, which is not the direct evidence of a binding between Mo and H₂O. 4) Line 287: "Under open-circuit condition, the rise of peak for H₂O and OH⁻ adsorption is detected, indicating that the exposed Mo atoms act as active sites for H₂O adsorption". Increase in the coordination number or FT intensity does not prove that this is the active site. The proposed mechanism, during which H₂O or OH enter in strained SVs of MoS₂ plane, cannot be only proved by shifts in the FT-EXAFS signals, because structural changes may be just induced by the applied potential.

Response: We sincerely thank you for these comments. For your convenience, we provide a point-by-point response to these comments:

1) Before we conducted the operando XAS measurements, we read previous literature about the operando XAS measurements and referred to the experimental design (Ref. 40). Owing to the limitation of test time at the beamline BL01C1, we can only choose two representative potentials for the operando XAS measurements. We also executed a careful and detailed discussion with beamline scientists. If the applied potential is too low, many important changes are difficult to detect. If the applied potential is too high, the huge amounts of hydrogen bubbles will interfere with the detection. The hydrogen evolution reaction was not dramatic at -0.05 V vs. RHE, allowing us to acquire valuable information while avoiding the interference of hydrogen bubbles. Besides, we choose -0.10 V versus RHE as a higher potential to probe the change trend of catalysts with the increase of potential.

Ref. 40: Cao, L. et al. Identification of single-atom active sites in carbon-based cobalt catalysts during electrocatalytic hydrogen evolution. *Nat. Catal.* **2**, 134-141 (2019).

2) We agree that it is not suitable to judge the probably binding of H₂O and OH only by the negative-shift of first shell. According to this comment, we superposed spectra

to compare the peak intensity, which were added in revised **Supplementary Information (Fig. R7)**. Besides, we also added the follow sentences to describe the spectra in revised manuscript:

In comparison with the ex situ condition, the main peak obtained under open-circuit condition displays a low- R shift, which is ascribed to the contribution of Ru-O bond (from the binding of H₂O and OH⁻) that overlapped with Ru-S bond. The contribution of Ru-O scattering also leads to the slight increase of the intensity of the main peak (Ref. 38, 40) (**Fig. R7**). During electrochemical H₂O reduction (-0.05 and -0.10 V vs. RHE), the peak shows a high- R shift by 0.07 Å. This indicates the distortion of coordination environment for Ru atoms, resulting from the redistribution of the electrons in Ru atoms between S ligands and the Ru-O bond (from adsorbed H₂O and OH⁻) under alkaline HER (Ref. 38, 40).

Ref. 38: Yang, H. B. et al. Atomically dispersed Ni(i) as the active site for electrochemical CO₂ reduction. *Nat. Energy* **3**, 140-147 (2018).

Ref. 40: Cao, L. et al. Identification of single-atom active sites in carbon-based cobalt catalysts during electrocatalytic hydrogen evolution. *Nat. Catal.* **2**, 134-141 (2019).

Fig. R7. Operando Ru K-edge FT-EXAFS spectra.

The FT-EXAFS spectra of Ru/np-MoS₂ recorded at different applied voltages.

3) We agree that the shift of rising edge indicates a change in the oxidation state, which is not the direct evidence of a binding between Mo and H₂O. For the previous literature (Ref. 38-40), the increase in the oxidation state of active sites under OCV

condition are general considered an indicator for the possible adsorption of reactant. In this work, we think that the enhancing water adsorption (including the adsorption on exposed Mo sites and Ru sites) need to be further determined by AP-XPS results (**Fig. 5**). According to this comment, we revised these sentences as following:

The rising edge of Ru/np-MoS₂ displays a positive-shift under open-circuit condition in relation to that under ex situ condition, meaning an increase of the Mo oxidation state (**Supplementary Figs. 28**). Different from np-MoS₂, the Mo sites in Ru/np-MoS₂ are exposed due to the formation of SVs. Thus, this change probably results from the binding of H₂O and OH⁻.

Ref. 38: Yang, H. B. et al. Atomically dispersed Ni(i) as the active site for electrochemical CO₂ reduction. *Nat. Energy* **3**, 140-147 (2018).

Ref. 39: Tian, X. et al. Engineering bunched Pt-Ni alloy nanocages for efficient oxygen reduction in practical fuel cells. *Science* **366**, 850-856 (2019).

Ref. 40: Cao, L. et al. Identification of single-atom active sites in carbon-based cobalt catalysts during electrocatalytic hydrogen evolution. *Nat. Catal.* **2**, 134-141 (2019).

4) We agree that the application of potential may induce some changes in catalysts. The catalysts always experience the reduction trend of the cathodic voltage under the operando HER measurement. In this work, the Ru sites in Ru/np-MoS₂ display the decrease in oxidation state when the HER occurring, resulting from that the adsorption of H₂O and OH⁻ cannot balance the reduction trend of the cathodic voltage. However, the exposed Mo sites in Ru/np-MoS₂ can balance the reduction trend of the cathodic voltage due to the abundant H₂O and OH⁻ adsorbed on exposed Mo sites without the subsequent dissociation, leading to the increase in oxidation state (Ref. 40).

Besides, we think that the changes in FT-EXAFS signals are not resulted from the application of potential. Because these changes of Ru/np-MoS₂ were not observed in np-MoS₂ which were operated under the same conditions. The difference between np-MoS₂ and Ru/np-MoS₂ is that the steric effect of Mo sites in Ru/np-MoS₂ is broken thus allowing the favorable charge interaction with H₂O. Therefore, these

changes are more likely caused by the adsorption behavior. It is true that the *operando* XAFS results cannot be used as the only means to verify the adsorption behavior. Therefore, we conducted the AP-XPS to probe the water adsorption capacity of Ru/np-MoS₂ and np-MoS₂. Fortunately, the results of AP-XPS are in good agreement with the theoretical prediction and *operando* XAS results, namely, the enhanced water adsorption capacity is main responsible for the improvement of catalytic activity.

Ref. 40: Cao, L. et al. Identification of single-atom active sites in carbon-based cobalt catalysts during electrocatalytic hydrogen evolution. *Nat. Catal.* **2**, 134-141 (2019).

Comment 12. It would be very useful if AP-XPS had been carried out under applied potential. A lack of this study is the absence of experiments performed at the S K-edge.

Response: We thank the reviewer for presenting these comments. Acquiring the AP-XPS data under applied potential is very help for further probe the adsorption behavior on the surface of catalysts. Unfortunately, the XPS device at 24A1 beamline of NSRRC does not support the in-situ electrochemical measurement and lack the related in-situ electrochemical cell at present. Meanwhile, since S K-edges locate at tender X-ray range (beamline BL16A1 of NSRRC), it is very difficult to acquire the useful information under *operando* condition. The spectral signal under *operando* is much weaker than that under *ex situ* condition. We tried many times, but did not get usable spectra. Many thanks for your understanding. Considering the difference between np-MoS₂ and Ru/np-MoS₂, we think that the isolated Ru sites and the exposed Mo sites may be the main active sites for water adsorption rather than the S sites. Therefore, we only focus on the Ru sites and Mo sites in this work.

Reviewer #2:

In this work, the authors have developed a strain engineering strategy to investigate the synergetic effect between single-atom Ru sites and SVs based on the Ru/np-MoS₂ sample. The successful introduction of strain is carefully demonstrated by adequate

structural and spectroscopy characterizations. To strengthen the understanding on the enhancement mechanism, the DFT calculations are combined with the operando XAS and ambient pressure XPS spectra to reveal the possible reaction process. Generally, the results are interesting and the findings here are attractive. However, there are still some scientific inconsistencies in the manuscript, which should be addressed before being further considered to publish in Nature Communications. Please find the detailed comments below.

Response: We appreciate the reviewer for your encouraging and constructive comments.

Comment 1. It is good that the Ru/np-MoS₂ has such a low Tafel slope of 31 mV dec⁻¹ in Figure 4e. Clearly, it will go through a Volmer-Tafel process rather than a Volmer-Heyrovsky process during HER, suggesting that two adsorbed H at the surface generated from water dissociation are coupled together to form H₂. However, it will never happen if the single Ru site is considered to complete the process alone. Thus, the current explanations on the DFT results is not consistent with the experimental results. If you look at the binding energy of H for S site (especially after applying the strain) in Figure 1d, it is a pretty good active site while it is ignored throughout the manuscript. If the Volmer-Tafel process is happening, it is highly possible that the H adsorbed at Ru are coupled with the H adsorbed at nearby S. The author has interpreted that the adsorbed H₂O in exposed Mo will dissociate at the Ru site. However, considering the abundance of H₂O in the electrolyte, as well as the more favorable adsorption energy on Ru site, such a prediction is unconvincing and confusing. What kind of MoS₂ are used for calculation, 1T or 2H?

Response: We sincerely thank you for these nice comments. According to these comments, we revised this part as following:

Simultaneously, Ru sites could easily activate H₂O molecule to generate intermediate H and OH species due to its lowest energy barriers of Volmer step. The subsequently H-H coupling can be completed by Ru sites and S sites, resulting from their low energy barriers of H* desorption.

We are sorry for our unclear statement. Actually, the role of exposed Mo sites is drag reactant (H_2O) into the inner Helmholtz plane, thus increase the abundance of H_2O in the inner Helmholtz plane. The similar behavior has been reported by the previous literature (Ref. 34). According to this comment, we have revised this sentence as following:

Owing to the high energy barrier of Volmer step for Mo_{SV} sites, the subsequent alkaline HER is blocked. Therefore, the Mo_{SV} sites may play a role of reactant (H_2O) dragging thus enhancing the mass transfer of H_2O to Ru sites.

According to the previous literature (Ref. 20, 23), the introduction of isolated metal atoms always leads to the 2H-1T phase transition. We think that the local region including isolated Ru atoms and the neighboring 1T- MoS_2 are the active region. Thus, we take 1T- MoS_2 as the model for the calculation. Our subsequent FT-EXAFS fitting curves of Ru/np- MoS_2 at Ru K-edge further identify the local structure of Ru/1T- MoS_2 (**Supplementary Fig. 14**). According to this comment, we added more details about the DFT calculation in **Supplementary Note 1**.

Ref. 34: Luo, Z. et al. Reactant friendly hydrogen evolution interface based on di-anionic MoS_2 surface. *Nat. Commun.* **11**, 1116 (2020).

Ref. 20: Wei, S. et al. Iridium-triggered phase transition of MoS_2 nanosheets boosts overall water splitting in alkaline media. *ACS Energy Lett.* **4**, 368-374, (2018).

Ref. 23: Luo, Z. et al. Chemically activating MoS_2 via spontaneous atomic palladium interfacial doping towards efficient hydrogen evolution. *Nat. Commun.* **9**, 2120 (2018).

Comment 2. In Figure 5c and 5e, the trend in the rising edge is not clear, please consider modify the layout of these figures. The information delivered from the insets is also ignored. The current explanations on these operando XANES data are a little awkward: If the Mo is really inert in np- MoS_2 , it needs to explain the shift in the data and why it is reduced. If the S is the site to bind with H_2O , it is unfair to be deduce from the results of Mo K-edge XANES. It has already been predicted that S sites are has more favorable binding energy for water dissociation and H-H coupling (Figure

1d). However, the important operando XAS for S edge are missing. Hypothetically, if the signals of Mo K-edge XAS spectra are true for Ru/np-MoS₂ in Figure 5e-f, it might not be a good sign. The obvious change due to the O species binding at the surface suggests the poisoning of the exposed Mo. Please make sure the data processing for Figure 4a, Figure 5d and 5f are the same and avoid the over-interpretation of these data.

Response: We appreciate you for this insightful and constructive recommendation.

According to your comments, we added the magnified rising edge XANES regions in revised **Supplementary Information (Supplementary Figure 28 / Fig. R8)** for the better comparison. The inset of **Fig. 5c, e** shows the first-order derivatives of the XANES spectra. The shift of rising edge will lead to the corresponding shift in the first-order derivative of the XANES spectrum. According to the comment, we added some sentences to further explain the first-order derivatives of the XANES spectra:

The rising edge of Ru/np-MoS₂ displays a positive-shift under open-circuit condition in relation to that under ex situ condition, meaning an increase of the Mo oxidation state (**Supplementary Figs. 28**). This is more obviously indicated by the first-order derivatives of the XANES spectra (inset of **Fig. 5e**).

Fig. R8. Magnified rising edge XANES regions.

Magnified rising edge XANES regions recorded at the Mo K-edge of np-MoS₂ and Ru/np-MoS₂.

For the Mo sites in np-MoS₂, the location of Mo sites (central sublayer) hinders the H₂O adsorption and dissociation due to the steric effect in np-MoS₂ (*Nat. Commun.*

10, 1217(2019)). Thus, it is difficult to explain the negative-shift of rising edge by the adsorption behavior of Mo sites. We think that the slight decrease in the oxidation state of Mo are resulted from the interaction between S atoms (outermost sublayer) and electrolyte, which leads to the redistribution of the electrons between Mo and S. The similar changes have been reported in Ref. 42.

According to the comments, we revised these sentences as following:

There is a negative-shift of rising edge under open-circuit condition compared with that under ex situ condition (**Supplementary Figs. 28 / Fig. R8**), indicating the decrease in the Mo oxidation state. It should be noted that the location of Mo sites (central sublayer) hinders the H₂O adsorption and dissociation due to the steric effect in np-MoS₂. Thus, the change of Mo oxidation state may result from the interaction between S atoms (outermost sublayer) and electrolyte (Ref. 42).

We agree that it is very useful to conduct in situ S K-edge XAS measures to further confirm the above conclusions. However, since S K-edges locate at tender X-ray range (beamline BL16A1 of NSRRC), it is very difficult to acquire the useful information under operando condition. The spectral signal under operando is much weaker than that under ex situ condition. We tried many times, but did not get usable spectra. Many thanks for your understanding. Considering the difference between np-MoS₂ and Ru/np-MoS₂, we think that the isolated Ru sites and the exposed Mo sites may be the main active sites for water adsorption rather than the S sites. Therefore, we only focus on the Ru sites and Mo sites in this work.

About the comment that O species binding at the surface suggests the poisoning of the exposed Mo. Using these changes in XANES and FT-EXAFS spectra as an indicator for the possible adsorption of reactant have been widely reported (Ref. 38-40). Meanwhile, we have revised many unclear descriptions in the revised manuscript (Page 14-15). For your convenience, we highlighted these parts with yellow color. Comparing the different behavior between Ru sites and Mo sites in Ru/np-MoS₂, the Ru sites can rapidly dissociate H₂O and complete the desorption of OH⁻. Thus, the Ru sites cannot balance the reduction trend of the cathodic voltage and show the decrease in oxidation state under alkaline HER. The exposed Mo sites can adsorb H₂O and OH⁻

but cannot dissociate reactant. This result in the accumulation of reactant, thus balancing the reduction trend of the cathodic voltage and shows the increase in oxidation state (Ref. 40). The accumulation of reactant on Mo sites will increase the concentration of reactants in the inner Helmholtz plane, which play an important role in the alkaline HER. This behavior has been widely reported (Ref. 34).

Ref. 42: Kornienko, N. et al. Operando spectroscopic analysis of an amorphous cobalt sulfide hydrogen evolution electrocatalyst. *J. Am. Chem. Soc.* **137**, 7448-7455 (2015).

Ref. 38: Yang, H. B. et al. Atomically dispersed Ni(i) as the active site for electrochemical CO₂ reduction. *Nat. Energy* **3**, 140-147 (2018).

Ref. 39: Tian, X. et al. Engineering bunched Pt-Ni alloy nanocages for efficient oxygen reduction in practical fuel cells. *Science* **366**, 850-856 (2019).

Ref. 40: Cao, L. et al. Identification of single-atom active sites in carbon-based cobalt catalysts during electrocatalytic hydrogen evolution. *Nat. Catal.* **2**, 134-141 (2019).

Ref. 34: Luo, Z. et al. Reactant friendly hydrogen evolution interface based on di-anionic MoS₂ surface. *Nat. Commun.* **11**, 1116 (2020).

Besides, we have carefully checked the data of Figure 4a, Figure 5d and 5f and make sure the data processing is the same.

Comment 3. About the oxidation state of Ru. The peak of the Ru/np-MoS₂ is shifted to higher binding energy for Ru 3p XPS spectrum in Figure 3a, while those for Mo 3d and S 2p in Supplementary Figure 14 are shifted oppositely. Thus, an electron injection should be responsible for the formation of 1T phase. Unfortunately, this issue is not included in the discussion. In this case, the Ru are losing electron and being oxidized? In the inset of Figure 5a, please provide more details about how to get the oxidation state of Ru (> 3+). Clearly, since the precursor to prepare Ru SACs is RuCl₃, the Ru are oxidized. It would be contradictory to the formation mechanism for Ru SACs and SVs (the reduction of Ru?). Please take good care of these judgements to avoid misunderstandings.

Response: We sincerely appreciate the reviewer for this reminding. We are sorry for these mistakes. After carefully check these data and the relevant literature (Ref. 23),

we have revised these sentences as following:

While the reduction of Mo by the electrons injection from Ru leads to the phase transformation of MoS₂ into the 1T structure, accompanied by the formation of SVs.

Besides, we also added some sentences to further explain the oxidation of Ru species:

The absorption-edge of Ru/np-MoS₂ locates between the RuCl₃ and RuO₂, suggesting that the oxidation of Ru species after doping in MoS₂. By injecting electrons from Ru species into the MoS₂ substrates, Mo species are reduced and cause phase conversion to form 1T-MoS₂, accompanied with the formation of SVs (Ref. 23).

Ref. 23: Luo, Z. et al. Chemically activating MoS₂ via spontaneous atomic palladium interfacial doping towards efficient hydrogen evolution. *Nat. Commun.* **9**, 2120 (2018).

About the determining of oxidation state of Ru, it can be roughly judged by the location of absorption-edge. The positive-shift of absorption-edge indicates the oxidation of Ru species. In the determining of oxidation state of Ru in **Fig. 5a**, we analyzed the absorption energy, which was obtained from the first maximum in the first-order derivative as the electron vacancy (Ref. 41). According to this comment, we revised these sentences and added **Supplementary Figs. 26 (Fig. R9)** for the further explanation:

To precisely determine the Ru valence state, the fitted oxidation states from the analysis of absorption energy are shown in the inset of **Fig. 5a** and **Supplementary Figs. 26**.

Fig. R9. The fitted average oxidation states.

The fitted average oxidation states of Ru from XANES spectra.

In the determining of oxidation state of Ru, we analyzed the absorption energy, which was obtained from the first maximum in the first-order derivative as the electron vacancy. The RuCl₃ (+3) and RuO₂ (+4) were used as the comparison standards.

Ref. 41: Kim, J. et al. High-Performance pyrochlore-type yttrium ruthenate electrocatalyst for oxygen evolution reaction in acidic media. *J. Am. Chem. Soc.* **139**, 12076-12083 (2017).

Comment 4. In Figure 3e, where does the 2H-MoS₂ come from? The comparison between the as-prepared np-MoS₂ samples and 2H-MoS₂ is not fair enough to illustrate the change in local bonding are due to the strain. If so, it needs to explain why there are contraction in Mo-S bond but extension in Mo-Mo bond.

Response: Thank you for the kind suggestion. The data of 2H-MoS₂ in **Fig. 3e** come from the previous literature (Ref. 24). We agree that the comparison between the as-prepared np-MoS₂ and 2H-MoS₂ is not fair. Thus, we delete these descriptions in the revised manuscript. In order to compare the change in local bonding, we prepared the plane MoS₂ and nanoporous MoS₂ with larger ligament with the same method except for changing the template (**Supplementary Note 2**). Then, we conducted the EXAFS spectra to probe the change in local bonding (**Fig. R5** and **Table R1**). The obtained EXAFS spectra show the that np-MoS₂ exhibits the greatest high-*R* shift of Mo-Mo peaks among these catalysts. The strain in these catalysts originated from the nanotube-shaped ligament thus formatting the atomically curved MoS₂. Therefore, the ligament with smaller diameter possess the most strained surface atom-arrangement, resulting in the extension in Mo-Mo bond (Ref. 21).

Fig. R5. FT-EXAFS fitting curves.

The FT-EXAFS spectra of np-MoS₂ in comparison with Lnp-MoS₂ and P-MoS₂. Corresponding FT-EXAFS fitting curves also shown in Fig. R5.

Table R1. Structural parameters extracted from the Mo K-edge EXAFS fitting.

Catalysts	Scattering pair	CN	R (Å)	σ^2 (10^{-3} Å ²)	ΔE_0 (eV)	R -factor
	Mo-Mo	3.9±0.5	3.149±0.01	2.48±0.8	-2.14	
Lnp-MoS ₂	Mo-S	5.9±0.5	2.405±0.01	2.63±0.6	2.53	0.008
	Mo-Mo	4.1±0.4	3.161±0.01	3.70±0.6	2.43	
np-MoS ₂	Mo-S	5.7±0.7	2.408±0.01	2.66±1.3	2.77	0.010
	Mo-Mo	4.1±0.5	3.169±0.01	3.42±1.3	2.90	

CN represents the coordination number; R represents the interatomic distance; σ^2 represents the Debye-Waller factor; ΔE_0 represents the edge-energy shift.

Ref. 21. Li, H. et al. Activating and optimizing MoS₂ basal planes for hydrogen evolution through the formation of strained sulphur vacancies. *Nat. Mater.* **15**, 364 (2016).

Comment 5. Surprisingly, the Ru content is actually very high (~8 at. %) for single-atom catalysts. Please make a more precisely comparison among literature and explain what happens in the as-prepared samples.

Response: We sincerely thank you for this nice suggestion. According to this suggestion, we added **Supplementary Figs. 18 (Fig. R10)** in revised **Supplementary Information** for further comparison and explanation:

The Ru content (~8 at%) in this work is relatively high compared with the works reported so far (**Supplementary Table 2**). The Ru/np-MoS₂ samples were prepared by a spontaneous reduction strategy reported by Ref. 23. Actually, Pd-MoS₂ with varied Pd contents (1-15% Pd-MoS₂) were prepared in Ref. 23. For the common synthesis strategy of single-atom catalysts, the number of defect or anchoring ligand always limit the metal load. However, the isolated metal atoms are substitutional doped into the MoS₂ by the spontaneous reduction strategy, thus avoiding the limitation of the number of defect or anchoring ligand. This is main responsible for the high metal content in the catalysts prepared by spontaneous reduction strategy.

Fig. R10. Characterizations of Ru content.

The compositions of Ru/np-MoS₂, Ru/P-MoS₂, and Ru/Lnp-MoS₂ result from EDS analyses.

Ref. 23: Luo, Z. et al. Chemically activating MoS₂ via spontaneous atomic palladium interfacial doping towards efficient hydrogen evolution. *Nat. Commun.* **9**, 2120 (2018).

Comment 6. Just for curiosity, how about the HER performance in acid? The authors try so hard to persuade that the strain can enhance the water adsorption properties of SVs and exposed Mo, it would be worthy to give it a shot at the acidic media, which might be helpful to support the claims that were made in this manuscript.

Response: Thank you for the kind suggestion. According to this comment, we added the acidic HER performance of catalysts in revised **Supplementary Information** to further emphasize the role of strain (**Fig. R11 / Supplementary Figure 22**). More details are shown in the response for *Comment 7*.

Comment 7. How to tell the impact on performance improvement due to 2H to 1T transition apart from induced strain?

Response: We are grateful to the reviewer for this nice comment.

In this work, by injecting electrons from Ru species into the MoS₂ substrates, Mo species are reduced and causes phase conversion to form 1T-MoS₂, accompanied with the formation of SVs (Ref. 23). In order to probe the impact on performance improvement due to 2H to 1T transition apart from induced strain, we performed the control experiment by comparing the ECSA-normalized current density of nanoporous MoS₂ and Ru doped MoS₂ under the same strain condition (Lnp-MoS₂ and Ru/Lnp-MoS₂ / np-MoS₂ and Ru/np-MoS₂) (**Supplementary Figure 22 / Fig. R11**). Obviously, both Ru/Lnp-MoS₂ and Ru/np-MoS₂ show the increase of current density as compared with Lnp-MoS₂ and np-MoS₂ due to the formation of local Ru/1T-MoS₂ active structure. Meanwhile, it is distinct that Ru/np-MoS₂ shows more increment of current density after the formation of local Ru/1T-MoS₂ active structure than that of Ru/Lnp-MoS₂. The phase transition in Ru doped MoS₂ results from the substitutional doping of Ru. By controlling the Ru content of Ru/np-MoS₂ and Ru/Lnp-MoS₂, the content of 1T-MoS₂ in Ru/Lnp-MoS₂ is less but very close to that of Ru/np-MoS₂ (**Supplementary Figure 23 / Fig. R12**). This indicates that the most strained Ru/1T-MoS₂ active structure in Ru/np-MoS₂ displays higher catalytic activity than less strained Ru/1T-MoS₂ active structure in Ru/Lnp-MoS₂, further highlighting the role of strain in boosting the catalytic activity of active structure.

Fig. R11. Catalytic HER performances.

Polarization curves of Ru/Lnp-MoS₂ and Ru/np-MoS₂ as compared with that of Lnp-MoS₂ and np-MoS₂ in alkaline (a) and acidic (d) solution. (b, e) Corresponding ECSA-normalized polarization curves. The increment of current density (at -0.10 V vs. RHE) after the formation of local Ru/1T-MoS₂ active structure for Ru/Lnp-MoS₂ and Ru/np-MoS₂ in alkaline (c) and acidic (f) solution. j represents the current density.

Fig. R12. XPS characterizations.

High-resolution Ru 3p (a), Mo 3d (b), and S 2p (c) XPS data of Ru/np-MoS₂ and Ru/Lnp-MoS₂.

Ref. 23: Luo, Z. et al. Chemically activating MoS₂ via spontaneous atomic palladium interfacial doping towards efficient hydrogen evolution. *Nat. Commun.* **9**, 2120 (2018).

Comment 8. The author has made great efforts to demonstrate the existence and

importance of strain. How about the impact of specific strain (such as pressure stress and tensile stress)? What kind of stress are taking about in this manuscript? How about the explorations of these stress in the literature?

Response: Thank you for the kind comment. Combining the previous literature (Ref. 21) and the morphology (nanotube-shaped ligament) of our catalysts, we think that the curved MoS₂ on the nanotube-shaped ligament are experience the bending stress (Fig. 4b / Fig. R13a). The resultant bending strain can be approximately replaced by the tensile strain at the atomic scale (as shown in Ref. 21) (Fig. 4c / Fig. R13b). This change can be detected by using the Mo-Mo radial distance as an indicator, as confirmed by the EXAFS results (Fig. 4a / Fig. R13c). By changing the size of ligaments of np-MoS₂, the tensile strain can be fine-tuned, thus regulating the atomic and electronic structures of catalysts. According to these comments, we added the above explanations in revised manuscript.

Fig. R13. characterizations of strain.

(a, b) Schematic of the atomic structure of Ru/Lnp-MoS₂ and Ru/np-MoS₂. The ε in (b) represents the amount of deformation. (c) The FT-EXAFS spectra of np-MoS₂ in comparison with Lnp-MoS₂ and P-MoS₂. Corresponding FT-EXAFS fitting curves also shown in Fig. R13c.

Ref. 21: Li, H. et al. Activating and optimizing MoS₂ basal planes for hydrogen evolution through the formation of strained sulphur vacancies. *Nat. Mater.* **15**, 364 (2016).

Reviewer #3:

This is a joint theory and experimental work on single atom catalysis, Ru doped in MoS₂. Finally, the importance of this work is to be judged based on what has been achieved experimentally. My comments, however, will be on the computational aspects. In my opinion, several crucial technical details about the DFT calculations have been left out, which hampers a full understanding, and reproduction if someone is interested, of the results and their validity. I am listing these below.

Response: Thank you for your positive comments on our manuscript.

Comment 1. The authors claim that they ‘hypothesized that the introduction of isolated Ru atoms into MoS₂ could cause the loss of S atoms around Ru atoms. It is not at all clear why such a hypothesis is physically reasonable. (Though they have shown subsequently that the experiments suggest so). Accepting that it is a reasonable hypothesis, an a posteriori validation could have been provided from DFT calculations. The authors have not done that. Or else, they can simply write that such a scenario was suggested by experimental observations. Since they have placed the theory first, making it the guide for subsequent experiments, a theoretical justification for such a hypothesis has to be presented.

Response: We sincerely appreciate the reviewer for this reminding. When we start this work, we read some previous literature (such as Ref. 20, 23) about metal doping into MoS₂ and find it may be a universal phenomenon that the doping of metal will result in the loss of S around the doped metal atoms. Especially, the Ref. 23 has made a comprehensive study on the formation of sulphur vacancies. Inspired by this, we established the model to conduct the theoretical prediction. We have cited these before the theoretical calculation. We agree that current hypothesis part is not reasonable. According to this comment, we added the follow sentence to further explain our hypothesis:

In light of previous reports regarding the activation of MoS₂ basal plane by the doping of isolated metal atoms (Ref. 23), we hypothesized that the introduction of isolated Ru atoms into MoS₂ could cause the loss of S atoms around Ru atoms, accompanied with phase conversion to form Ru/1T-MoS₂. The formation of SVs could break the steric

effect and allow the direct binding between Mo atoms and H₂O molecule in SVs. This hypothesis was suggested by experimental observations in Ref. 23.

Ref. 20: Wei, S. et al. Iridium-triggered phase transition of MoS₂ nanosheets boosts overall water splitting in alkaline media. *ACS Energy Lett.* **4**, 368-374, (2018).

Ref. 23: Luo, Z. et al. Chemically activating MoS₂ via spontaneous atomic palladium interfacial doping towards efficient hydrogen evolution. *Nat. Commun.* **9**, 2120 (2018).

Comment 2. On the same page they write ‘Ru sites of Ru/MoS₂ possess much lower water adsorption energy of -0.516 eV as compared to that of Mo_{SV} sites ...’ and refer to Fig. 1(b) and Supplementary Fig1. Now Fig. 1 only shows the structure, gives no quantitative information about water adsorption. Supplementary Fig. 1 shows calculated water adsorption energies. The final argument about efficacy of Ru sites over Mo_{SV} in H₂O adsorption, dissociation and subsequent HER is based on the free energies presented in Fig 1(d), which is the correct way of looking at it. How are the free energies derived from the energies presented in Suppl. Fig 1? It is important to tell the reader how the zero-point energy and entropy contributions are calculated or obtained.

Response: We thank the reviewer for these nice comments and questions. We have revised these wrong annotations in the revised manuscript. According to these comments, we added some explanations (**Supplementary Note 1** in revised **Supplementary Information**) to further explain our calculation part:

The generally accepted alkaline HER mechanism consists of two steps, with the Volmer step followed by the Tafel step or Heyrovsky step:

The free energies of step (1) and step (3) should be the same at equilibrium potential. Computations on the exact free energy of OH⁻ in solutions could be avoided by using computational hydrogen electrode based on the above assumption (Ref. 7).

The free energies of steps 1-3 are calculated as the following equation:

$$\Delta G = \Delta E + \Delta ZPE - T\Delta S \quad (4)$$

Where, ΔG , ΔE , ΔZPE , and $T\Delta S$ are the changes for free energy, enthalpy from DFT calculations, zero-point energy and entropy (at 300 K), respectively. ΔZPE is derived after frequency calculation by the following equation (Ref. 8):

$$ZPE = \frac{1}{2} \sum hv_i \quad (5)$$

Where h is the Planck constant, v_i are the computed vibrational frequencies.

The TS values of adsorbed species are calculated with the vibrational frequencies, as shown in following equation (Ref. 9):

$$TSv_i = k_B T \left[\sum_i \ln \left(\frac{1}{1 - e^{-hv_i/k_B T}} \right) + \sum_i \frac{hv_i}{k_B T} \frac{1}{(e^{hv_i/k_B T} + 1)} \right] \quad (6)$$

Where k_B is the Boltzmann constant, T is the temperature.

Ref. 7 in Supplementary Information: Blöchl, P. E. Projector augmented-wave method. *Phys. Rev. B* **50**, 17953 (1994).

Ref. 8 in Supplementary Information: Zheng, Y. et al. High electrocatalytic hydrogen evolution activity of an anomalous ruthenium catalyst. *J. Am. Chem. Soc.* **138**, 16174-16181 (2016).

Ref. 9 in Supplementary Information: Nørskov, J. K. et al. Trends in the exchange current for hydrogen evolution. *J. Electrochem. Soc.* **152**, J23-J26 (2005).

Comment 3. Neither Fig 1(d) nor Suppl. Fig 1 tells the amount or the nature of the strain.

Response: We sincerely thank you for this comment. According to your comment, we have added the nature of the strain in **Supplementary Note 1**:

For the application of strain, our preliminary work before this work suggests that the np-MoS₂ experience tensile strain (about 10%) originated from the nanotube-shaped ligament (Ref. 5, 6). Ideally, the uniaxial tensile strain (10%) was applied on the above model. The subsequent HAADF-STEM characterizations show that the value of tensile strain is about 12% (**Fig. 2g**).

Ref. 5 in Supplementary Information: Tan, Y. et al. Monolayer MoS₂ films supported by 3D nanoporous metals for high-efficiency electrocatalytic hydrogen production. *Adv. Mater.* **26**, 8023-8028 (2014).

Ref. 6 in Supplementary Information: Chen, D. et al. General synthesis of nanoporous 2D metal compounds with 3D bicontinuous structure. *Adv. Mater.* **32**, 2004055 (2020).

Comment 4. The authors write ‘Even if the energy barrier of water dissociation for Mo_{SV} site slightly decreases after the applied strain, its value still ...’ The free energies reported in Fig 1(d), by themselves, do not give the barriers for H₂O dissociation or the HER process, in my opinion. These are the free energies of some intermediate steps in the whole process. There may be (generally are) further kinetic barriers in between. The authors must clarify this, re-write this part so as not to overstate the results.

Response: Many thanks for the comments. We are sorry for the wrong description in this part. According to these comments, we revised this part as following:

Even if the energy barrier of Volmer step for Mo_{SV} site slightly decreases after the applied strain, its value still much higher than that of Ru site (**Fig. 1d**), suggesting the sluggish Volmer step on Mo site.

Meanwhile, we revised the similar descriptions in this part. For your convenience, we highlighted these descriptions with yellow color.

Comment 5. They also write within the section on theoretical results ‘The subsequent H-H coupling step also highlights the role of strain ...’ How? I do not see this addressed within DFT.

Response: We thank the reviewer for presenting this nice question. We are sorry for the unclear description. According to this comment, we revised this description as following:

Besides, the application of strain also decreases energy barrier of H* desorption for Ru sites and S sites, thus leading to the enhanced ability for H-H coupling.

Comment 6. Details about what system they have taken for DFT calculations have not been reported. These are customary. Is it a single layer of MoS₂ or a nanotube? What is the size? What is the size of the vacuum layer in the non-periodic direction? What are the formation energies of Ru substitution, S vacancy etc.? Such details must be reported in the Suppl. Info.

Response: We appreciate you for this valuable recommendation.

In this work, a unit cell (4×4×1) of 1T-MoS₂ was select to establish the model. During the DFT calculations, a vacuum slab of 20 Å was added along the surface of 1T-MoS₂. According to these comments, we added this information in the **Supplementary Note 1**. Besides, we added the calculation of formation energy in **Supplementary Fig. 1 (Fig. R14)**:

Fig. R14. The calculation of formation energy.

(a) Formation energy of S-vacancy in 1T-MoS₂. (b) 1) Formation energy of Ru atom replaces Mo site. 2) Formation energy of S-vacancy in Ru/1T-MoS₂

The formation energy (E_f) of Ru atom replaces Mo site was calculated as following equation:

$$E_f = E_{doped} - E_{perfect} + (\mu_{Mo} - \mu_{Ru})$$

Where, E_{doped} , $E_{perfect}$, μ_{Mo} , and μ_{Ru} are the total energy for MoS₂ with Ru doped, the total energy for perfect MoS₂, the chemical potential for Mo, and the chemical potential for Ru, respectively.

The formation energy (E_f) for S-vacancy in 1T-MoS₂ was calculated as following equation:

$$E_f = E_{SV} + \mu_S - E_{perfect}$$

Where, E_{SV} , μ_S , and $E_{perfect}$ are the total energy for MoS₂ with S-vacancy, the chemical potential for S, and the total energy for perfect MoS₂, respectively.

The formation energy (E_f) for S-vacancy in Ru/1T-MoS₂ was calculated as following equation:

$$E_f = E_{Ru/SV} + \mu_S - E_{Ru/perfect}$$

Where, $E_{Ru/SV}$, μ_S , and $E_{Ru/perfect}$ are the total energy for Ru/1T-MoS₂ with S-vacancy, the chemical potential for single S atom, and the total energy for perfect Ru/1T-MoS₂, respectively.

We also added some sentences in the revised manuscript to further explain our calculations:

The formation energy of Ru atom replacing Mo site was calculated (**Supplementary Fig. 1**). It is shown that Ru exhibits a tendency to replace Mo with an exothermic energy of -0.650 eV, indicating the substitutional doping of Ru is a thermodynamically-driven process. Then, we calculated the formation energy of SVs in 1T-MoS₂ and Ru/1T-MoS₂, which show the decrease in the formation energy of SVs by 0.832 eV after Ru doping, proving the feasibility of using Ru doping to create SVs.

Comment 7. As I said, the main validation of the work has to come from the experimental part. But, in addition, the above shortcomings in the computational part have to be addressed before it becomes suitable for publication.

Response: We appreciate you for these insightful and valuable recommendation.

REVIEWER COMMENTS

Reviewer #1 (Remarks to the Author):

Some points of this manuscript may still remain questionable, but I think that this is the maximum that can be achieved from these data and calculations. The authors have addressed the points in a satisfactory manner, and I then recommend the publication of the manuscript.

Reviewer #2 (Remarks to the Author):

After the major revision, this version can be considered to be published without further revision.

Reviewer #3 (Remarks to the Author):

In the revised manuscript, the authors have attempted to address the shortcomings I had pointed out in their previous manuscript. While some aspects have become clear now, I think there are still some issues that need correction.

The authors write

1. 'The projector augmented wave (PAW) method was conducted to describe the inert core electrons.' They perhaps mean 'to describe/treat interactions between the valence electrons and the ion cores.'
2. 'A vacuum slab of 20 Å was added along the surface.' Perpendicular to the surface?
3. 'The unit cell was optimized until ...' Was the unit cell optimized? Which means optimizing its size and shape. Or were the atom positions optimized? This point must be made clear.
4. 'Besides, the application of strain also decreases energy barriers of H* desorption for Ru sites and S sites, thus leading to the enhanced ability for H-H coupling.' What the authors mean by H* desorption barrier, and how that is obtained from DFT calculations is still not clear to me. H* desorption barrier involves kinetic processes, which have not been (and are not easy) treated here.
5. ΔG is a change of entropy. Authors state how they calculated entropy of the adsorbed species. How is the entropy of the free species calculated? Exactly what free species have they considered?
6. Is there a typo Eqn 6 in SI?
7. What are the reference states for Mo, Ru and S in formations energy calculations? These must be stated for others to be able to understand and reproduce the results.
8. Supplementary Fig 2 still states 'Adsorption Energy'. Which quantity is this— ΔG , ΔE , or something else? In other places they talk about enthalpy?
9. Do the DFT results give enthalpy or energy?

I think these details have to be corrected/clarified before the manuscript can be published.

Responses to the Referees' Comments

We thank the referees for their valuable comments and positive endorsement to our manuscript. We have carefully considered the referees' comments and revised the manuscript accordingly. Our responses and corresponding revisions are as follows:

Reviewer #1:

Some points of this manuscript may still remain questionable, but I think that this is the maximum that can be achieved from these data and calculations. The authors have addressed the points in a satisfactory manner, and I then recommend the publication of the manuscript.

Response: We appreciate your recommendation of acceptance and helpful comments in the reviewing process and are pleased to have our manuscript be reviewed by you.

Reviewer #2:

After the major revision, this version can be considered to be published without further revision.

Response: We are very grateful to your encouraging and positive comments and really appreciate your agreement of acceptance with this revised manuscript.

Reviewer #3:

In the revised manuscript, the authors have attempted to address the shortcomings I had pointed out in their previous manuscript. While some aspects have become clear now, I think there are still some issues that need correction.

Response: Many thanks for your positive comments on our manuscript. We have revised our manuscript accordingly.

Comment 1. 'The projector augmented wave (PAW) method was conducted to describe the inert core electrons.' They perhaps mean 'to describe/treat interactions between the valence electrons and the ion cores.'

Response: We appreciate the reviewer for this reminding. We have revised this sentence as following:

The projector augmented wave (PAW) method was conducted to describe/treat interactions between the valence electrons and the ion cores.

Comment 2. ‘A vacuum slab of 20 Å was added along the surface.’ Perpendicular to the surface?

Response: We sincerely thank you for this comment. We have revised this sentence as following:

A vacuum layer of 20 Å was added perpendicular to the slab surface.

Comment 3. ‘The unit cell was optimized until ...’ Was the unit cell optimized? Which means optimizing its size and shape. Or were the atom positions optimized? This point must be made clear.

Response: Thank you for the kind questions and suggestions. Usually, the adatom or dopant cannot change the lattice structure of the MoS₂, but causing local distortions. Thus, while we optimize the structures, we keep the lattice structure stable and optimize the positions of atoms. We have added these sentences in Supplementary Note 1 of **Supplementary Information**.

Comment 4. ‘Besides, the application of strain also decreases energy barriers of H* desorption for Ru sites and S sites, thus leading to the enhanced ability for H-H coupling.’ What the authors mean by H* desorption barrier, and how that is obtained from DFT calculations is still not clear to me. H* desorption barrier involves kinetic processes, which have not been (and are not easy) treated here.

Response: We sincerely appreciate the reviewer for this reminding. We are sorry for this mistake. We have revised this sentence as following:

Besides, the application of strain also decreases the hydrogen adsorption free energy for Ru sites and S sites, thus leading to the enhanced ability for H-H coupling.

The calculations of hydrogen adsorption free energy were shown in Supplementary

Note 1 of **Supplementary Information**.

Comment 5. Delta G is a change of entropy. Authors state how they calculated entropy of the adsorbed species. How is the entropy of the free species calculated? Exactly what free species have they considered?

Response: We thank the reviewer for the insightful comment. In the DFT calculation, we did not consider the free H₂O molecules in solutions. The calculations of free energies for each step of HER general consider the adsorbed species (such as Ref. 24). The free energies for Volmer step and Heyrovsky step should be the same at HER equilibrium potential. Computations on the exact free energy of free OH⁻ in solutions could be avoided by using computational hydrogen electrode based on the above assumption (Ref. 24 / Ref. 8 in **Supplementary Information**). We agree that taking the free H₂O molecules into account can more truly reflect the actual catalytic environment. We will perform more comprehensive and detailed calculations in the follow-up work. Many thanks for your understanding.

Ref. 24: Huang, Y. et al. Atomically engineering activation sites onto metallic 1T-MoS₂ catalysts for enhanced electrochemical hydrogen evolution. *Nat. Commun.* **10**, 982 (2019).

Ref. 8 in **Supplementary Information**: Blöchl, P. E. Projector augmented-wave method. *Phys. Rev. B* **50**, 17953 (1994).

Comment 6. Is there a typo Eqn 6 in SI?

Response: We appreciate the reviewer for this reminding. We are sorry to make such a mistake. The correct equation is shown as following:

$$TS_v = k_B T \left[\sum_K \ln \left(\frac{1}{1 - e^{-hv/k_B T}} \right) + \sum_K \frac{hv}{k_B T} \frac{1}{(e^{hv/k_B T} - 1)} + 1 \right]$$

Comment 7. What are the reference states for Mo, Ru and S in formations energy calculations? These must be stated for others to be able to understand and reproduce the results.

Response: We sincerely thank you for this nice question. In the formation energy calculations, we need to calculate the chemical potential of Mo, Ru, and S atoms. In this paper, the calculated chemical potentials are equal to the DFT total energies of their ground states (Ref. 1 in **Supplementary Information**). We have added these sentences in Supplementary Fig. 1 of **Supplementary Information**.

Ref. 1 in **Supplementary Information**: Emery, A. A. & Wolverton, C. High-throughput DFT calculations of formation energy, stability and oxygen vacancy formation energy of ABO₃ perovskites. *Sci. Data* **4**, 170153 (2017).

Comment 8. Supplementary Fig 2 still states ‘Adsorption Energy’. Which quantity is this- ΔG , ΔE , or something else? In other places they talk about enthalpy?

Response: We are grateful to the reviewer for this nice comment. Supplementary Fig. 2 display the calculations of H₂O adsorption energy ($\Delta E_{\text{H}_2\text{O}}$) at different sites. The $\Delta E_{\text{H}_2\text{O}}$ is used to evaluate the water affinity of active sites (*Nat. Commun.* **9**, 2533 (2018) | DOI: 10.1038/s41467-018-04954-7). Fig. 1d display the free energies (ΔG) of active sites on the surface of catalysts. In order to further clarify our calculations, we added more explanations in revised manuscript (Page 5). For your convenience, we show all changes in the manuscript with colour highlighting.

Comment 9. Do the DFT results give enthalpy or energy?

Response: Thank you for the kind question. All the DFT calculations were performed on the VASP software. Current, the VASP software cannot give the results of the enthalpy (H) directly. Usually, it can calculate the energy of different structures, such as MoS₂, Ru-doped MoS₂, and MoS₂ with adsorbed species. Also, the vibrational frequency and entropy (S) of the systems can be calculated directly. While all the data are ready, we calculate the adsorption energy, the value of TS (T=298.15 K in VASP) and the zero-point energy (E_{ZPE}), and the Gibbs free energy is further calculated.

I think these details have to be corrected/clarified before the manuscript can be published.

Response: We truly thank you for reviewing the revised version of our manuscript and greatly appreciate your helpful comments.

REVIEWERS' COMMENTS

Reviewer #3 (Remarks to the Author):

The authors have addressed all of my concerns except one. Before going to that, I would like to point out that I do not find Eqn 6 in the Supplementary Information in the latest version. Am I missing something?

Now the concerns. Going by the 'corrected' form of the expression for TS the authors have given in the response to referees, there are a few questions.

1. What does the ν signify on the left hand side?

2. What is the index K being summed over?

3. Most importantly, how do they get the +1 (last term in the square brackets)? What is its origin? They should check this expression carefully. There is a potential problem if this form has been used to calculate TS. To convince the readers that all quantities have been calculated correctly, I suggest they share the calculated frequencies, ZPE values, entropies, adsorption energies and free energies (just as done in Ref. 24 they have cited) in the SI.

Responses to the Referees' Comments:

Reviewer #3:

The authors have addressed all of my concerns except one. Before going to that, I would like to point out that I do not find Eqn 6 in the Supplementary Information in the latest version. Am I missing something?

Response: Many thanks for the kind question. The Eqn 6 (Eqn 7 in revised **Supplementary Information**) was shown in page 34 of **Supplementary Information**. We have added new tags (Equation 7) to make it easy to find these equations quickly.

Now the concerns. Going by the 'corrected' form of the expression for TS the authors have given in the response to referees, there are a few questions.

Comment 1. What does the ν signify on the left hand side?

Response: We sincerely thank you for the comments about Equation 7 (Calculations of TS). After comprehensive consideration of your suggestions, we revised the Equation 7 and provided the detailed derivation as following:

The TS values of adsorbed species are calculated in the following equation (Ref. 11):

$$TS^{vib}(T, \nu) = \sum \kappa T \left(\frac{\beta h \nu_i}{\exp(\beta h \nu_i) - 1} - \ln(1 - \exp(-\beta h \nu_i)) \right) \quad (\text{Equation 7})$$

Where $\beta = 1/\kappa T$, $\{\nu_i\}$ are vibrational modes, κ is the Boltzmann constant, T is the temperature (which is set to 298 K in the present work), respectively. The detailed derivation can be found in Ref.11 in **Supplementary Information**.

Ref. 11 in **Supplementary Information**: 14. Reuter, K. & Scheffler, M. Composition, structure, and stability of RuO₂ (110) as a function of oxygen pressure. *Phys. Rev. B* **65**, 035406 (2001).

Comment 2. What is the index K being summed over?

Response: We appreciate you for this question. The detailed of Equation 7 were provided in **Comment 1**.

Comment 3. Most importantly, how do they get the +1 (last term in the square brackets)? What is its origin? They should check this expression carefully. There is a potential problem if this form has been used to calculate TS. To convince the readers that all quantities have been calculated correctly, I suggest they share the calculated frequencies, ZPE values, entropies, adsorption energies and free energies (just as done in Ref. 24 they have cited) in the SI.

Response: We appreciate the reviewer for these nice questions and suggestions. After careful examination of the Equation 7, we revised the Equation 7 and provided the origins of Equation 7 (Please see *Comment 1*). Besides, we shared the calculated frequencies, ZPE values, entropies, adsorption energies and free energies in the **Supplementary Information** as following:

Supplementary Table 1. Key values for water dissociation and hydrogen generation on the surface of different models.

Models	$\Delta E(\text{H}_2\text{O})/\text{eV}$	$\Delta ZPE(\text{H}_2\text{O})+U_{(0 \rightarrow \text{T})}(\text{H}_2\text{O})/\text{eV}$	$\Delta TS(\text{H}_2\text{O})/\text{eV}$	$\Delta G(\text{H}_2\text{O})/\text{eV}$
Ru/MoS ₂	0.295	0.590+0.102	0.236	0.751
Strained Ru/MoS ₂	-0.154	0.602+0.064	0.128	0.384
Models	$\Delta E(\text{H}^*)/\text{eV}$	$\Delta ZPE(\text{H}^*)+U_{(0 \rightarrow \text{T})}(\text{H}^*)/\text{eV}$	$\Delta TS(\text{H}^*)/\text{eV}$	$\Delta G(\text{H}^*)/\text{eV}$
Ru sites	0.557	0.224+0.010	0.013	0.778
Mo sites	0.596	0.267+0.005	0.006	0.862
S sites	0.945	0.208+0.013	0.018	1.148
Strained Ru sites	-0.035	0.227+0.008	0.011	0.189
Strained Mo sites	0.651	0.175+0.008	0.012	0.822
Strained S sites	0.459	0.211+0.012	0.017	0.665